# Fault zone heterogeneities explain depth-dependent pattern and evolution of slow earthquakes in Cascadia

Yingdi Luo [1,2 ✉] & Zhen Liu [2]

Slow earthquakes including tremor and slow-slip events are recent additions to the conventional earthquake family and have a close link to megathrust earthquakes. Slow earthquakes along the Cascadia subduction zone display a diverse behavior at different spatiotemporal scales and an intriguing increase of events frequency with depth. However, what causes such variability, especially the depth-dependent behavior is not well understood. Here we build on a heterogeneous asperities-in-matrix fault model that incorporates differential pore pressure in a rate-and-state friction framework to investigate the underlying processes of the observed episodic tremor and slow-slip (ETS) variability. We find that the variations of effective normal stress (pore pressure) is one important factor in controlling ETS behavior. Our model reproduces the full complexity of ETS patterns and the depth-frequency scaling that agree quantitatively well with observations, suggesting that fault zone heterogeneities can be one viable mechanism to explain a broad spectrum of transient fault behaviors.

[1] JIFRESSE, University of California, Los Angeles, CA, USA. [2] Jet Propulsion Laboratory, California Institute of Technology, Pasadena, CA, USA. ✉email: luoyingd@jpl.nasa.gov

Recent advances in seismology and geodesy have led to many new findings of unconventional slow earthquakes such as slow-slip event (SSE) and non-volcanic tremor (NVT). SSE, mainly discovered through geodetic measurements, has much slower slip rate (centimeters to meters per year) and slow propagation speed (kilometers per day) compared to regular earthquakes, and does not emit detectable seismic signals. NVT is known for its seismic yet weak, non-impulsive nature. SSE and tremor are mostly found at young subduction zones around the world[1], and typically locate adjacent to seismogenic zones capable of generating devastating megathrust earthquakes. Despite slow or weak signature, SSE and NVT are shown to play an important role in earthquake dynamics and hazard estimation due to its potential link to megathrust earthquakes[2,3]. SSE and NVT activities are sensitive to stress perturbations due to their low stress drop nature[4–7]. Stress perturbations from earthquakes or other sources can impact SSE occurrence[3,8,9]. The broad-scale nucleation processes of megathrust earthquakes[10] has been suggested to affect nearby SSE and tremor patterns, making them potential candidates for precursory monitoring of future large earthquakes[3].

Cascadia subduction zone (CSZ) is one of the closely monitored subduction zone faults due to its potential for magnitude 9 megathrust earthquake and catastrophic tsunami hazard that can affect populated urban centers along its thousand-kilometer-long margin (e.g., M9 project, https://hazards.uw.edu/geology/m9). Tremor and SSE in CSZ are often spatially and temporally correlated and referred to as episodic tremor and slow-slip (ETS)[11–14]. ETS events in CSZ mostly occur in the seismic-aseismic transition zone (thereafter referred to as ETS zone) between the shallow seismogenic and deep creeping regions over a depth range of ~25–55 km. ETS in CSZ displays clear along-strike variation with northern, middle, and southern segments[15] (Fig. 1). In general, a hierarchical spectrum of ETS activities with various sizes and depth has been observed (Figs. 1 and 2). These ETS phenomena can be categorized into several essential types: (1) "Major ETS." This refers to large ETS events that tend to occur semi-regularly and can propagate through one or multiple segments up to several hundred kilometers (Figs. 1 and 2). Major ETS events usually propagate at a relatively steady speed of several kilometers per day bilaterally along strike with one dominant branch. Occasionally, they propagate unilaterally or collide toward each other (Supplementary Figs. 1 and 2). Major ETS typically nucleate at mid-deep portion of ETS zone (>38 km depth) then gradually expand up-dip and fill in the entire ETS zone. These events have a recurrence interval of approximately 13 months and 2 years in the northern and middle segments, and a less regular interval averaging ~7 months in the southern segment. (2) "Deep ETS" (indicated by blue inverted triangles in Fig. 1, see also Fig. 2). This refers to smaller episodes of ETS, propagating along strike with similar speed as major ETS but over a shorter distance of only a few tens of kilometers. Unlike the major ETS, deep ETS events are usually confined to the mid-deep part of ETS zone. (3) "Deep arrested ETS" (Fig. 2). From time to time ETS event also nucleates in the deep ETS zone (approximately below 46 km) but localizes in a small region of about 10–30 km in size and fails to develop into a propagating ETS event. Such ETS event is analogous to the arrested earthquakes with regard to regular earthquakes[16]. (4) "Individual tremor" (Fig. 2 and Supplementary Figs. 1 and 2). This refers to the isolated tremor occurring individually without developing into any kind of previously defined ETS types, which is also commonly referred to as low-frequency earthquakes (LFE)[17]. Individual tremor is very common in the southern segment, sometimes seen in the northern segment but rarely observed in the middle segment. Here the depth boundaries of ~38 and ~46 km are used

as the first-order approximation to characterize the ETS types at depth.

Most studies so far have focused on the major ETS and regard other types as background[18–21]. Here we extend the term of ETS to all sustained tremor activities including deep and arrested ETS. Even though the associated slow-slip has not been observed for smaller ETS events[22], it is likely due to geodetic observation limit, and smaller ETS events share the same mechanism as the major ones especially with respect to the origination of ETS. In fact, we find that the major, deep, and arrested ETS are very similar in the early phase, as they usually nucleate at the deep part of ETS zone and propagates upwards[23], showing no apparent difference until cascading into larger size. Overall, ETS shows an increase of frequency (decrease of recurrence interval) with depth along the entire subduction margin (Fig. 1b–d, see also[4]). Typically, only the larger ETS can propagate into an area at a shallow depth while the smaller arrested ETS events occur almost exclusively at the deeper portion of the ETS zone (Figs. 1 and 2). Such depth-dependent behaviors for slow earthquakes were also observed in other subduction zone such as southwest Japan[24]. The hierarchical ETS pattern displays considerable similarities to regular earthquakes. For instance, individual tremor activities and propagating ETS events over a larger area can find their counterparts in conventional earthquake family as microseismicity/seismic swarm and a sizable earthquake (see more with "Partial Instability" and "Total Instability" in Luo and Ampuero[25]). Similarly, the deep ETS events can occur in the form of arrested or propagating, and occasionally cascade into major ETS events, just like the self-arresting earthquakes evolving into runaway earthquakes[16,26]. One standing question is whether the ETS is controlled by the same physical mechanism as regular earthquakes. Recent studies suggest that slow earthquakes in CSZ follow the similar scaling law as regular earthquakes and display pulse-like ruptures similar to seismic ruptures[20].

Despite the improved observations, our understanding of ETS behaviors and underlying mechanics is still limited. In particular, what physical models can self-consistently explain the hierarchical ETS distribution and evolution? Over the past decade, significant modeling efforts have been made with attempts to capture some characteristics of ETS. Some aim to reproduce tremor character and certain scaling relation, but not actual spatiotemporal patterns and associated SSE[27]. Others simulate various SSE behaviors without accompanying tremor[3,28–30]. The third group consists of models being able to reproduce ETS with both SSE and tremor. Early efforts attempt to reproduce ETS without prescribing any physical tremor asperities[31–33]. Later the asperity-in-matrix (AIM) model, supported by the geological observations of exhumed subduction fault zone materials comprised of competent phacoides embedded in incompetent background matrix[34,35], was proposed to consider actual tremor "asperities" (small seismogenic patches representing fault heterogeneities) from which tremors naturally originate. The AIM models have shown great potential for reproducing rich phenomena of slow earthquakes in Japan[36,37]. More recently, one-dimensional (1-D) AIM model was developed to successfully reproduce some notable ETS features in CSZ such as the large-scale forward migration and rapid tremor reversals (RTRs)[38]. Nevertheless, so far no model is able to explain the full spectrum of observed ETS complexity, especially the depth-dependent ETS pattern and the relation and evolution of small to large ETS.

We hypothesize that the along-dip variations of the differential pore pressure (thus effective normal stress) of the asperities and/or the background matrix can be one plausible candidate for the observed along-dip ETS variability in CSZ. This hypothesis is motivated by the strong correlation between ETS behaviors and stress condition as shown by previous studies[4,7,25,33,39,40], and a

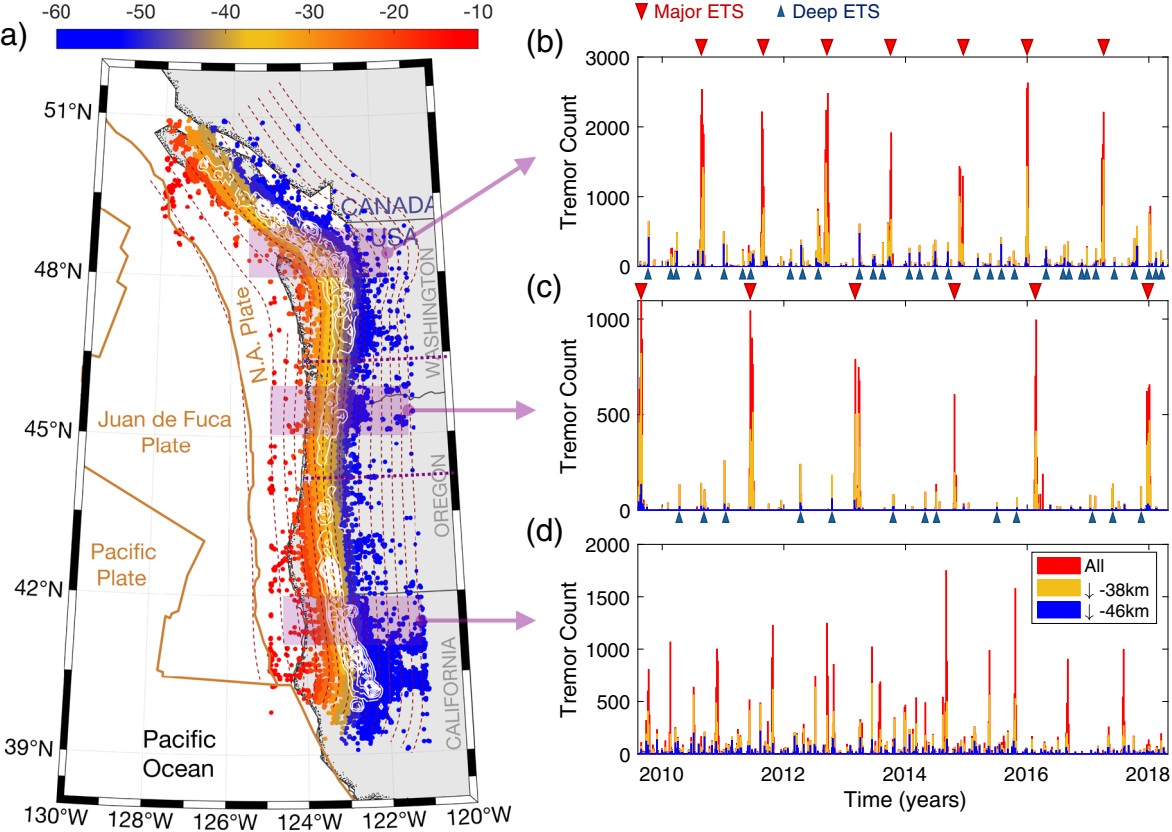

**Fig. 1 Space-time variation of tremor in Cascadia subduction zone (CSZ). a** Map view of tremor distribution from Pacific Northwest Seismic Network (PNSN) tremor catalog (https://pnsn.org) of August 2009 to April 2018 along CSZ. Colored circles are tremor locations, color coded with inferred tremor depth (km) assuming they are on the plate interface. White contour shows normalized cumulative tremor density distribution. Brown curves are plate boundaries from PB2002[70]. Dark-red dashed lines indicate 5-km iso-depth contours of plate interface[71]. **b–d** Binned (5-day) tremor activity as a function of time of selected area from the three episodic tremor and slow-slip (ETS) segments (indicated as purple shades in (**a**)), color coded with depth. Red, yellow, and blue color indicate the tremor activities of the whole fault, below 38 km depth (mid) and 46 km depth (deep), respectively. Note that the depth boundary of 38 and 46 km used here is not definite yet presents a good approximation for ETS categorization: **b** 48–49°N in northern segment; **c** 45–46°N in middle segment; **d** 41–42° in southern segment. Examples of major and deep ETS events in the northern and middle segments are indicated with red and blue triangles, respectively. Note that the smaller ETS (arrested) almost exclusively occur at the deeper (blue) portion of the ETS zone.

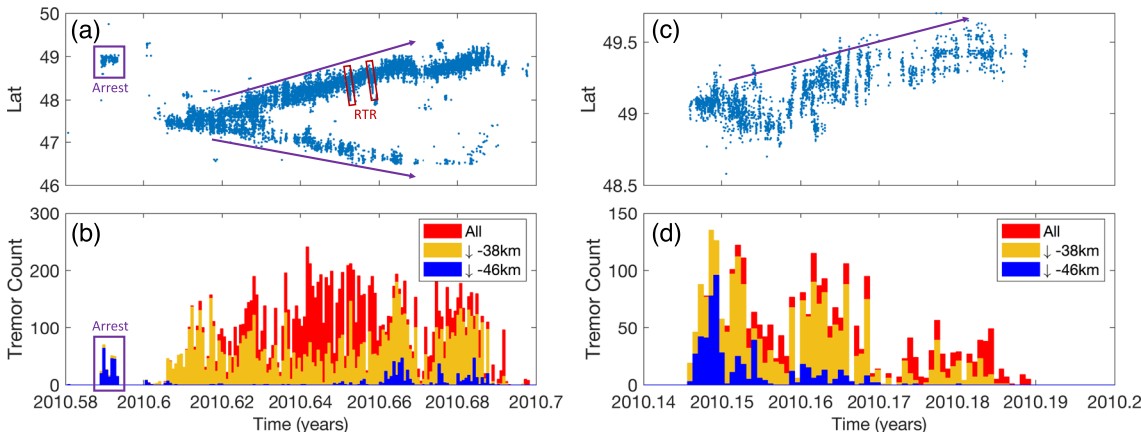

**Fig. 2 Spatiotemporal distribution of episodic tremor and slow-slip (ETS) in the northern segment of Cascadia subduction zone. a, b** show an example of "major ETS" in 2010. **a** Tremor catalog as a function of time and latitude. **b** Corresponding tremor activity binned in 6-hour interval, similar to Fig. 1b–d. The "major ETS" nucleates at mid-depth range and propagates both along-strike (indicated with purple arrows in (**a**)) and up-dip (color-coded distribution in (**b**)), with significant tremor activities at shallow depth after the initial phase. Purple boxes show the localized (arrested) deep ETS ("deep arrested ETS"). Examples of rapid tremor reversals (RTRs) are indicated with red boxes. **c, d** An example of "deep ETS." Figure convention follows **a, b**.

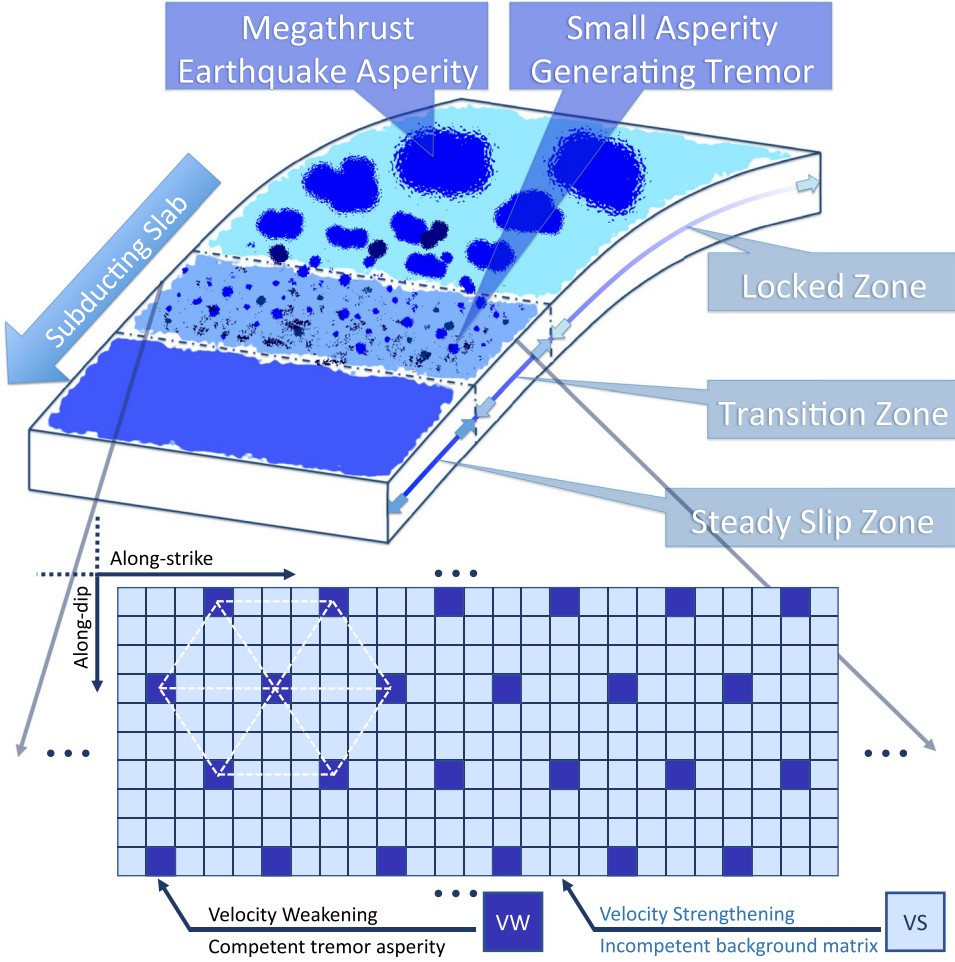

**Fig. 3 Schematic view of heterogeneous subduction zone.** Concept model of subduction zone fault showing downdip seismic-aseismic transition zone (episodic tremor and slow-slip "ETS" zone) with tremor asperities (dark-colored) embedded in the background matrix (light-colored). (Zoom-in) The ETS transition zone is discretized as 2-D fault with regularly spaced single-cell velocity-weakening (VW, dark-colored) tremor asperities embedded in the velocity-strengthening (VS, light-colored) background.

conceptual model[41] that suggests temperature-dependent healing and permeability reduction in silica-rich fault gouge via dissolution–precipitation creep could lead to the decrease of overall effective normal stress with depth, affecting the depth-dependent ETS phenomena. Here we combine the geological, seismological constraints in CSZ, and develop a 3-dimensional (3-D) rate-and-state friction AIM model to investigate the complex spatiotemporal ETS patterns. We show by considering depth-dependent variations of differential pore pressure following a simple linear profile, our model is able to reproduce the full spectrum of the observed ETS complexity including depth-dependent ETS distributions in Cascadia. These results provide important insights into the origination and evolution of ETS and suggest possible connection between slow and regular earthquakes.

## Results

**Asperity-in-matrix models reproducing episodic tremor and slow-slip events.** We incorporate 3-D complexity into our previous 1-D along-strike AIM model by taking into account the depth-dependent frictional properties and stress conditions. The fundamentals of rate-and-state friction and technical model details can be found in the Methods section. Our model consists of a planar 2-D fault embedded in 3-D elastic medium. The ETS zone is represented by heterogeneous velocity-weakening (VW)

tremor asperities embedded within velocity-strengthening (VS) matrix, as illustrated in Fig. 3. We consider three end-member scenarios: (1) a reference model without depth-dependent effective normal stress variation; (2) a model with a sharp transition with depth in effective normal stress; and (3) a model with gradual decrease in depth-dependent effective normal stress. Thereafter, we refer the three models as uniform, bi-modular, and linear models, respectively (Fig. 4 and Supplementary Figs. 3 and 4).

The results of the three models are summarized in Fig. 5. The uniform model without depth-dependent effective normal stress generates only characteristic major ETS events with a recurrence interval of approximately 1.5 years and very little background tremor in between (Fig. 5a, see also Supplementary Movie 1). Despite the model having asperities with random criticalness, simulated major ETS events have well-defined propagation patterns. The ETS event tends to fill the entire depth range of the ETS zone and mostly propagates unilaterally with a propagation speed of a few kilometers per day (Supplementary Fig. 5a). The major ETS sometimes nucleates at the shallow part of the fault likely related to the uniform stress condition. The bi-modular model featuring a relatively high stress zone in the shallow-half and a low stress zone in the deep-half reproduces only bi-modular ETS (Fig. 5b, see also Supplementary Movie 2): major and deep ETS. The major ETS propagates through the

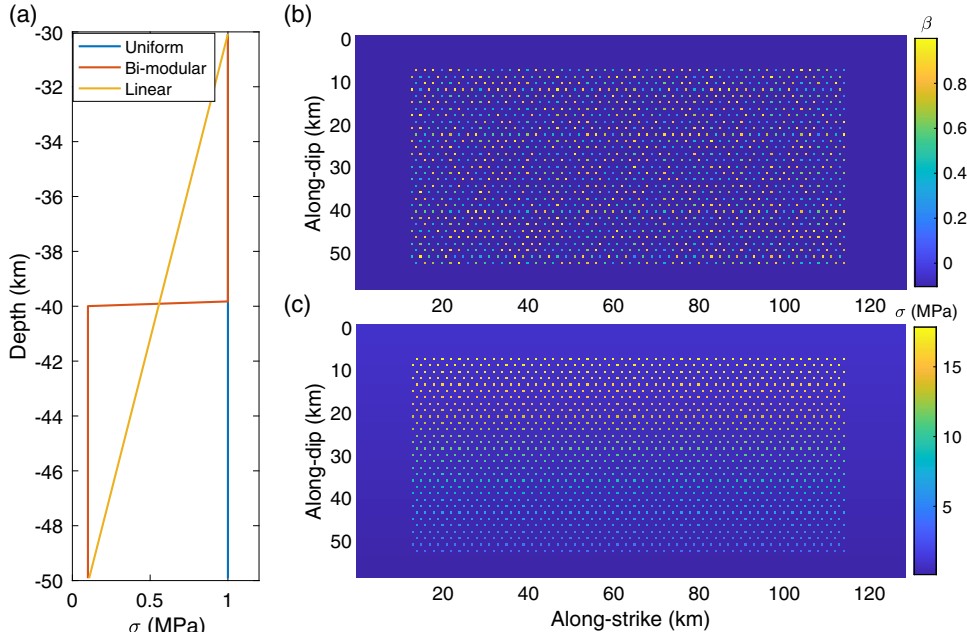

**Fig. 4 Rate-and-state model setup. a** The along-depth profile of effective normal stress (σ) of the background matrix for three end-member models considered: uniform model (blue), bi-modular model (red), and linear model (yellow). **b** Fault normal view of asperity criticalness $\beta = L_{asp}/L_c$ (asperity size / critical length). **c** Effective normal stress σ in the linear model.

entire fault of both the shallow and deep ETS zone, with ~1.5 years recurrence interval, similar to the uniform model (Supplementary Fig. 5b), while deep ETS is mostly confined to the deep low stress zone with a recurrence interval of 2–3 months (Supplementary Fig. 5c). Note the burst of tremor activities resulting faster back-propagating NVTs at deeper part during the major ETS event resembles the RTRs observed in Cascadia[42]. The linear model, as a natural extension of spatially averaged bi-modular model, has a linear decrease of effective normal stress with depth. We find that simulated major ETS events in the linear model have a slightly shorter interval of around 1.2 years. The linear model not only features a more natural stress profile, but also reproduces a much richer set of ETS events with various magnitudes and depth (Fig. 5, see also Supplementary Movie 3). All these are in good agreement with observations in Cascadia.

**Depth-dependent stress distribution leads to hierarchical ETS.** We further examine the long-term ETS behavior in the linear model by extending the simulation time to 40 years. We also increase the output (snapshot) frequency so more tremor activities can be recorded during each ETS event. The parameters and initial conditions are based on the result of the first 4 years simulation reported previously. The results are summarized in Figs. 5 and 6 (see also Supplementary Fig. 5d and Supplementary Movie 4). Figure 5d–f shows the 40 years simulated tremor activities projected along the strike and dip direction and the binned tremor activities. The linear model is able to reproduce strikingly similar spatiotemporal ETS patterns as compared to the real-world observations in northern Cascadia (Fig. 1b). In particular, all essential observed ETS types (major, deep, arrested, and individual tremor) are reproduced with this linear model. Figure 6a–c (see also Fig. 7a–j) shows a typical example of simulated major ETS event that lasts about 25 days. This ETS event nucleates at a deeper part of the fault due to plate loading and low stress, then propagates bilaterally along strike at a speed of approximately 4~5 km per day, with one branch of higher tremor activities than the other. The event also progressively propagates up-dip during the entire episode. A total of 37 major

ETS events are produced by our model during the 40 years of simulation, yielding an average recurrence interval of around 13 months, a close match to the observed major ETS recurrence interval in the northern segment of the CSZ (Fig. 8). The moment magnitude $M_w$ (see details in Methods section) of simulated major ETS is in the range of 6.0–6.4, bounded by the along-strike dimension of our model (128 km). Extrapolating to a 500 km fault suggests an upper bound of $M_w \sim 6.8$ in agreement with observations[19,22,43]. Note that albeit the linear model presented here is tailored to reproduce ETS patterns in northern segment, the model can be easily adjusted for other segments. The spatiotemporal variation (Fig. 5d–f) of each ETS episode shows good agreement with observations (Fig. 1b and Supplementary Fig. 1). In particular, the modeled ETS intervals display an exponential distribution with depth that agree very well with the observed depth-frequency scaling (Fig. 8, see also Figure 4 of Weck and Creager[4]). Most simulated ETS events propagate along strike either bilaterally with a dominant branch or nearly unilaterally probably due to heterogeneous material properties and stress residuals from previous events. We also noticed some interesting variations of major ETS, e.g., the "colliding" ETS in which two ETS branches nucleated at similar time and propagates toward each other and eventually collide (Supplementary Fig. 5d). Such colliding behavior has been observed in Cascadia (Supplementary Fig. 2). The model also reproduces multiple deep ETS events. Figure 6d–f (see also Fig. 7k–p) shows a typical example of modeled deep ETS event. Such event nucleates at deep part of the ETS zone, propagating along strike and expanding up-dip generally. However, it is not able to propagate into the shallow ETS region due to the stress barrier induced by previous major ETS events, which have larger stress drop than the deep ETS event due to depth-dependent stress condition. In the case when the stress barrier is eventually dissipated, the ETS event will develop into a major one. Due to even lower stress condition (closer to lithostatic), deep ETS is usually weaker and more frequent than the major ETS event. In our rate-and-state model the deep ETS events are interspersed between major ETS events with a varying recurrence interval of several months, comparable to the

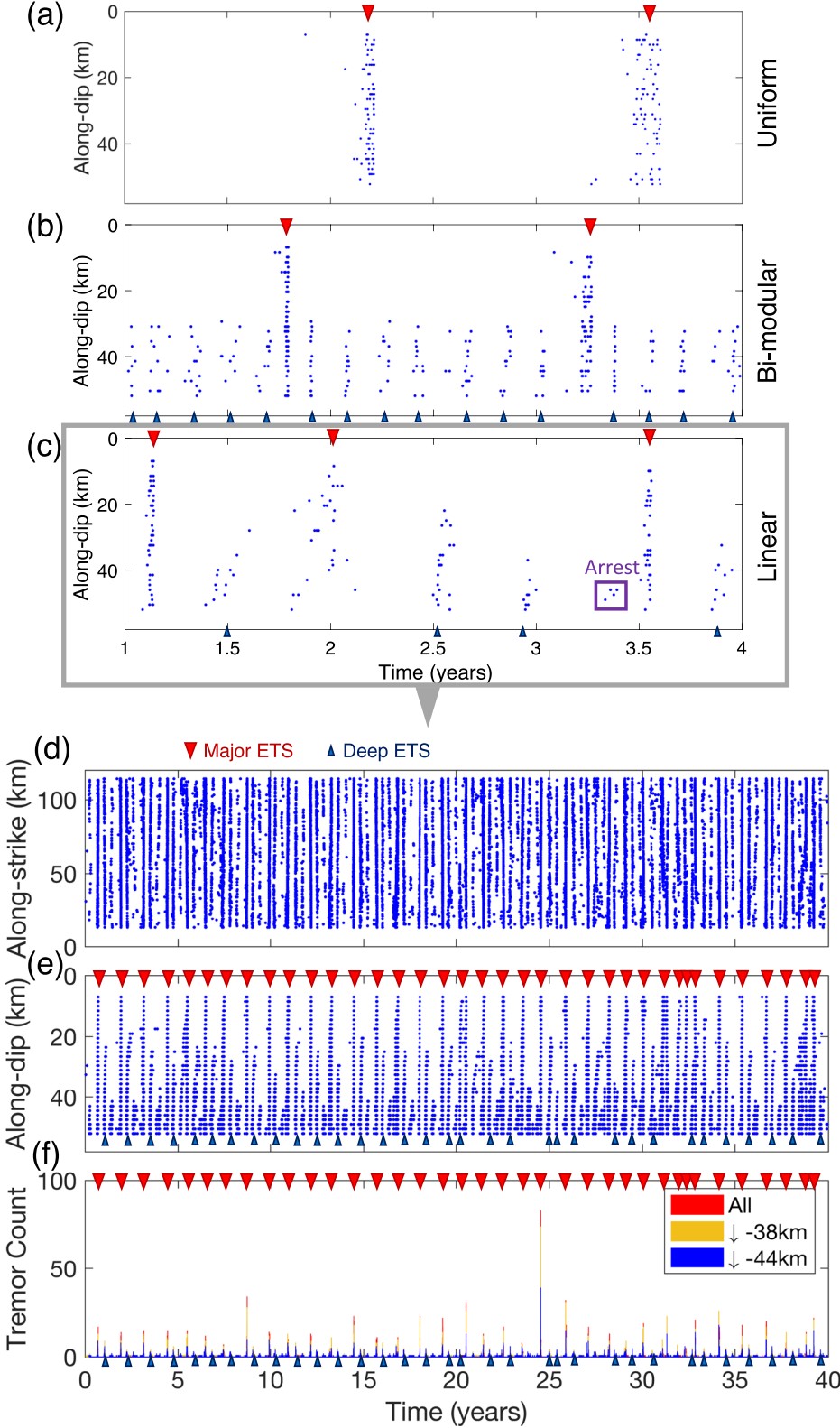

**Fig. 5 Rate-and-state model results. a–c** Simulated tremor activities from three asperity-in-matrix (AIM) models with slip-rate detection threshold of $10^4 V_{pl}$ (tectonic loading rate). **a** Uniform model, **b** bi-modular model, and **c** linear model. **d–f** Forty years simulation results of the linear model with **d** strike- and **e** dip-parallel projection, **f** tremor activities binned in 5-day window. Major ETS (episodic tremor and slow-slip, red triangles) with approximately 13 months recurrence and interspersed deep and arrested ETS (blue triangles) with interval of months. The spatiotemporal patterns of simulated ETS are in strikingly good agreements with the observations in northern Cascadia (e.g., Fig. 1b), see also Fig. 8.

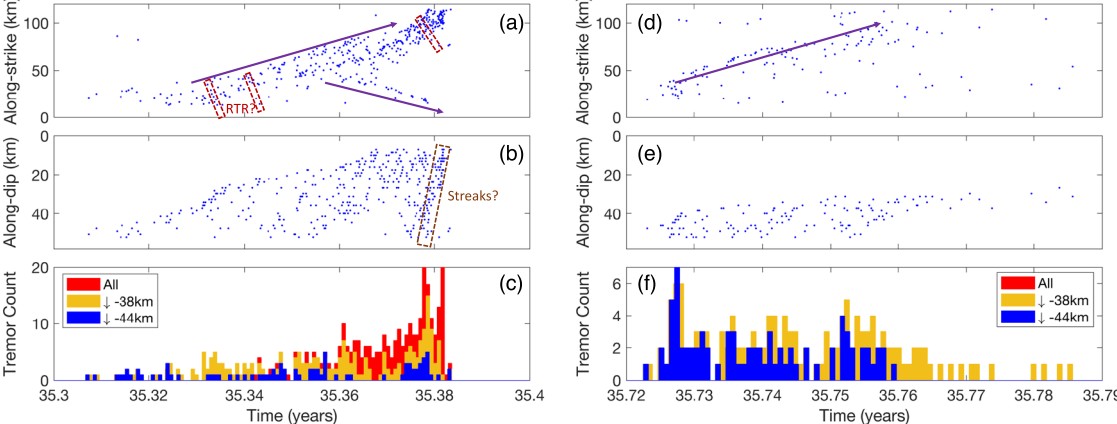

**Fig. 6 Modeled episodic tremor and slow-slip (ETS) variability.** Similar to Fig. 2 with additional plot of seismicity projected to along-dip direction (**b**, **e**). **a–c** Zoomed-in view of a major ETS event. Bottom panel shows tremor activities binned in 6-h window. The ETS nucleates at deep part and propagates bilaterally along-strike with rupture speed of approximately 4~5 km per day, with one branch of more tremor activities than the other, and the ETS progressively propagates up-dip. Examples of possible rapid tremor reversals (RTRs) are indicated with red dashed boxes. Note also the plausible along-dip fast "tremor streaks" (indicated with brown dashed box) toward the end of the ETS as suggested by[42,46]. **d–f** Zoomed-in view of a deep ETS, which stays mostly in the deep portion of the fault, with a general trend of up-dip propagation.

observations (Figs. 1b–d and 8). Our model also captures the deep arrested ETS that is localized without developing into propagating ETS. Figure 7q–s shows a typical example of deep arrested ETS that nucleates at the deep portion of the ETS zone and grows (semi-)circularly but eventually dies out and fails to develop into a larger event. The deep arrested ETS is very similar to a deep ETS or major ETS in terms of nucleation/early phase. Without the knowledge of rupture history, one usually cannot tell the difference until it develops into a larger event. The single ETS (individual tremor) in our model is basically background tremor, i.e., isolated NVT event (LFE) takes place individually and sporadically without forming into an episode (e.g., Fig. 7t). Estimated $M_w$ of the deep ETS is in the range of ~4.5–6.0, in line with observations[22].

In summary, for the first time, our depth-dependent AIM model with linear stress variation is able to reproduce the full hierarchy of ETS behaviors, ranging from small individual tremor, to arrested and runaway deep ETS, to the characteristic major ETS, as well as the exponential depth-frequency scaling, all in good quantitative agreement with the spatiotemporal ETS patterns observed in Cascadia. Our model suggests that fault zone heterogeneities in terms of frictional properties and effective normal stress as a result of differential pore pressure is a viable mechanism underlying the broad spectrum of the ETS variabilities in CSZ.

**Discussion**
In this study, we develop a 3-D rate-and-state AIM model with simple linear depth-dependent stress profile and employ it in multi-cycle simulations to investigate the hierarchical ETS in CSZ. Our model is constrained by geological, geodetic, and seismological observations and inspired by previous conceptual and numerical models. To our best knowledge, this physically sound model is the first in the community that reproduces the full spectrum of observed tremor and slow-slip complexity in Cascadia with the potential applicability to slow earthquake processes worldwide.

Our model results suggest that the overall effective normal stress of the ETS zone can be as low as several to a fraction of MPa, consistent with near-lithostatic pore-fluid pressure in the ETS zone as a result of fluids sealing due to low gouge permeability[44]. The low effective stress is a necessary condition

for slow rupture processes such as SSE and relatively short recurrence interval of ETS, making them susceptible to the variation in stress conditions. Previous studies show tremor and SSE can be modulated by small stress perturbation such as tidal stress[7,39,45] or tectonic stressing[3] and reference therein. Our model quantitatively demonstrates that the change of ETS frequency and size with depth is a result of a systematic decrease of effective normal stress. Such change of effective normal stress with depth could result from the temperature- (thus depth-) dependence of healing and gouge permeability reduction due to progressive silica deposition in overlying forearc crust, leading to fluid overpressure buildup and reduced effective fault normal stress[41]. Our model highlights the need to explicitly consider the resultant stress conditions as controlled by forearc structure in modeling ETS and megathrust faulting behaviors.

In our bi-modular and linear model, we observed that ETS consistently nucleates from the deeper part of ETS zone in agreement with real-world observations[4], while ETS nucleation in the uniform model does not have clear preference despite the same loading conditions. We attribute the realistic ETS nucleation in both bi-modular and linear model to the joint contribution of lower effective normal stress (thus material strength) and continuous loading at the deeper part of ETS zone. As the depth dependence of effective normal stress results in the relative shear strength decrease with depth, tremor asperities at depth tend to be weaker and fail more often than asperities up-dip. It is possible that the shear strength decrease with depth could also associate with friction decrease as a function of depth[4]. However, lack of constraints on frictional parameters makes it difficult further testing this interpretation. Instead, our AIM model demonstrates that the depth variation of effective normal stress can be one major controlling factor affecting fault zone strength and ETS variations, supported by available geological and seismic observations[41].

In addition to the depth dependence of fault zone properties, other factors in our model such as the asperity distribution and their interaction can also contribute to the ETS variability and evolution. For example, RTRs seem to favor the low stress region of ETS zone (e.g., Fig. 7 and Supplementary Fig. 5). The asperities of relatively high stress drop can have stronger interactions with neighboring asperities, especially with those of relatively low stress drop, resulting in cascading reactivation of already ruptured

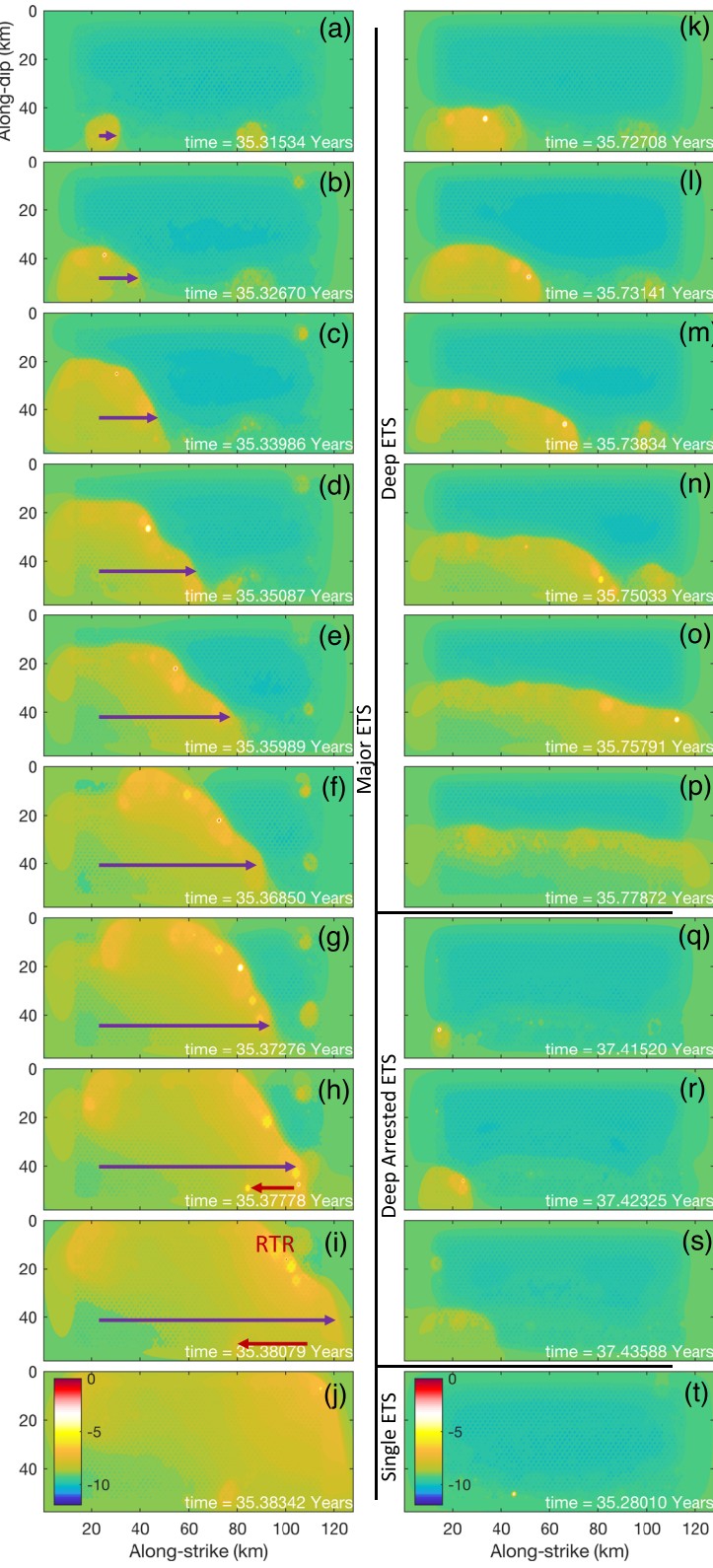

**Fig. 7 Snapshots of modeled episodic tremor and slow-slip (ETS) variability.** Color shows logarithmic slip rate ($V$ : m · s$^{-1}$): blue, $V <$ tectonic loading rate $V_{pl}$ (locked); turquoise: $V \simeq V_{pl}$ (creeping around plate loading rate); yellow: slow-slip; white-red: tremor (seismic, $V > 10^4 V_{pl}$). Purple arrow shows ETS forward migration. Red arrow shows a plausible example of rapid tremor reversal. Snapshots of **a**–**j** time evolution of a major ETS, **k**–**p** a deep ETS, **q**–**s** a deep arrested ETS, and **t** individual tremor.

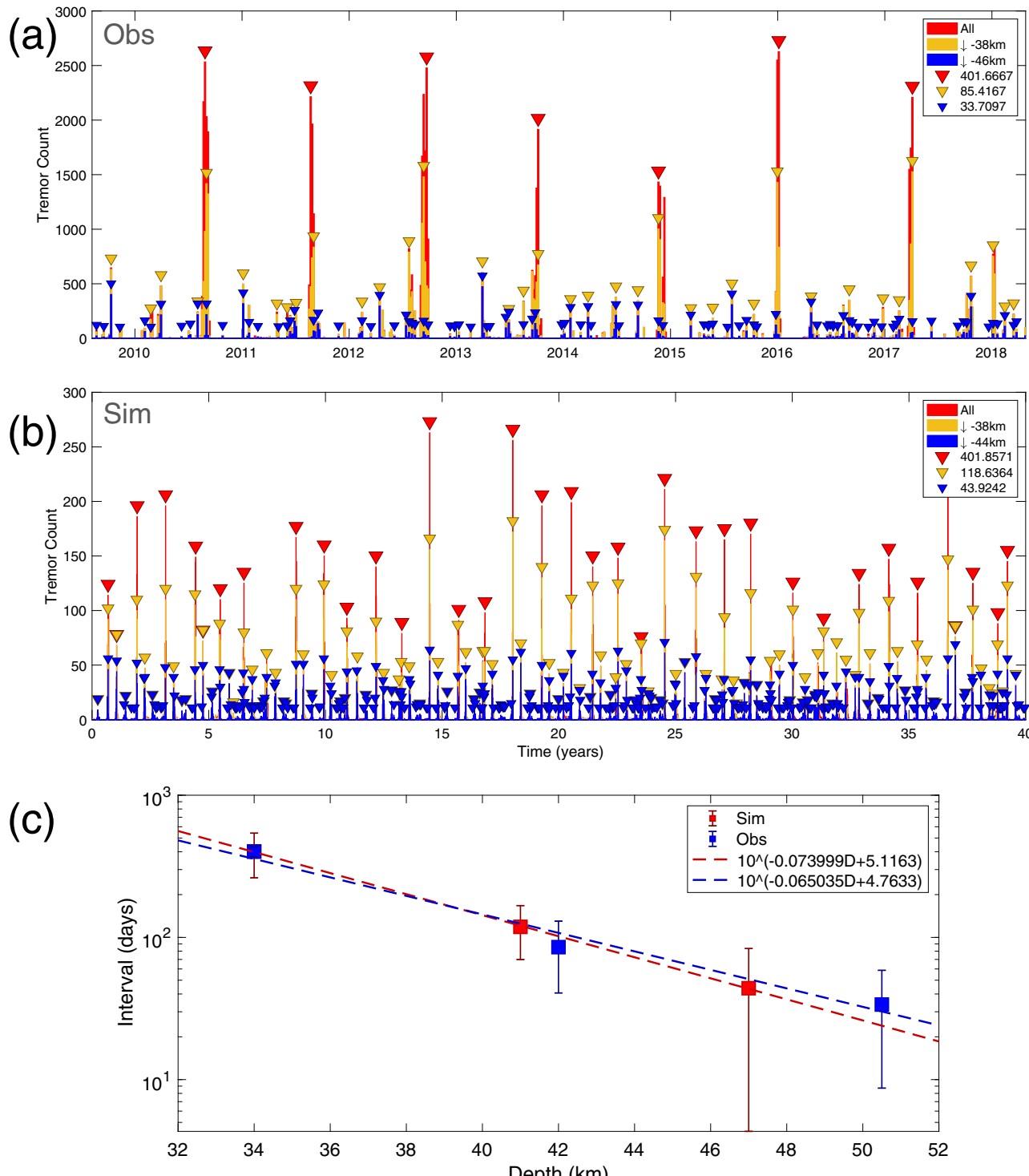

**Fig. 8 Comparison of the spatiotemporal distribution of episodic tremor and slow-slip (ETS) between observation and rate-and-state model.**
**a** Observations over the period of August 2009–April 2018. Similar to Fig. 1b, red, yellow, and blue triangles indicate automatically picked ETS events at shallow (<38 km), mid (38–46 km), and deep (>46 km) depth. The average event intervals are approximately 401.7, 85.4, and 33.7 days, respectively.
**b** Results from the linear rate-and-state model in 40-year simulation. ETS events at shallow (<38 km), mid (38–44 km), and deep (>44 km) depth have average intervals of 401.9, 118.6, and 43.9 days, respectively. **c** The ETS interval-depth (center depth of each ETS bin) relations from observations (blue) and model (red) follow a strikingly similar exponential distribution (error bar = 1 standard deviation, dash-lines: linear regression).

neighboring asperities and develop into local tremor burst. This is most pronounced in the bi-modular case, that the asperities failed in the high stress region can reactivate the asperities in the low stress region, causing the RTRs to be most active near the boundary of high and low stress regions (Supplementary Fig. 5b

and Supplementary Movie 2). Assisted by elevated background slip rate associated with ongoing ETS, RTRs are also frequently observed at transient stress heterogeneity boundary due to the integrated effect of asperity interaction, plate loading, and residual stress from ETS history. This suggests that the locations of

such dynamic tremor bursts, for instance, RTRs and tremor streaks[46–48] might indicate a high degree of stress heterogeneity contributing to cascading interaction of tremor asperities and resultant tremor swarm. In this model, transient stress heterogeneities as a result of heterogeneous stress drop of asperities from past ETS history can sometimes couple with local variation of asperity strength to cause tremor-less slip[23], in which reduced slip rate is sustained without incurring seismic asperity rupture.

Our AIM-based model also reveals some good similarities between hierarchical ETS patterns and ordinary earthquakes. For example, the individual tremor and ETS as compared to microseismicity and seismic swarm, arrested and runaway ETS as compared to arrested and runaway earthquakes. During the major ETS, slip reactivation is common (e.g., RTRs, Supplementary Fig. 5b, and Supplementary Movie 2), similar to slip reactivation observed in megathrust earthquake[49]. Slow earthquakes were originally thought to have different scaling relations from conventional earthquakes (linear vs. cubic moment-duration relation[5]). But more recently slow earthquakes are found to actually follow the similar scaling relation as conventional earthquakes[22,50]. The similarities between slow and regular earthquakes and their similar scaling laws suggest that they might have a close connection and share the similar dynamic properties and mechanism. Luo and Ampuero[25] use a simplified two-degree-of-freedom model to demonstrate that slow earthquakes and regular earthquakes are transformable depending on the evolution of stress condition (Fig. 9). If the simplified model can be extended, then the heterogeneous AIM fault model with differential stress conditions presented here will have broad implications toward explaining the full spectrum of earthquake faulting processes.

The possible transformability between slow and regular earthquakes and the capability of smaller ETS evolving into larger ETS events raise one natural question: could a major ETS develop into an even larger ETS that penetrates further up-dip beyond the current inferred ETS region? In extreme case, could Cascadia ETS cascade into a major megathrust earthquake similar to the 1700 M9 earthquake? Long-term SSEs have been found at shallower depth in Japan and other subduction zones. Decades long observations in CSZ show no sign of such long-term ETS. While there has been no major earthquake in CSZ so far with modern instrumentation, observations of slow-to-fast earthquake interplay worldwide might shed some light on it. For instance, recent research[10] showed that large-scale aseismic slow-slip occurs as a prelude to megathrust earthquakes. Also SSEs adjacent to megathrust seismogenic zone may display noticeable spatiotemporal pattern changes before large earthquake due to the interaction between seismogenic and ETS zone[3]. In this regard the spatiotemporal pattern changes of Cascadia ETS might serve as potential candidate to highlight precursory change prior to a major megathrust earthquake. Testing this would require incorporation of both megathrust earthquake region and ETS region (Fig. 3) with realistic fault zone properties and geometric settings in the AIM model, which we plan to address in a separate study. Nevertheless, the promising results from this study demonstrate the capability of AIM ETS model in explaining a wide range of complex slow earthquake phenomena and support its application to other subduction zones around the globe, as supported by geological and other observational constraints.

## Methods

**Rate-and-state model and fault stability**. The laboratory-derived rate-and-state friction[51–53] has been widely used for modeling slip transients from laboratory to natural earthquake scales[54]. Under rate-and-friction tectonic fault is treated as a frictional surface, such that the fault is assumed to be always slipping. With the "quasi-dynamic" approximation, fault shear stress $\tau$ is equal to the product of effective normal stress $\sigma$ (defined as fault normal stress minus pore pressure) and fault friction $\mu$

$$\sigma\mu = \tau = \tau_{\text{trans}} + \tau_{\text{per}} - \tau_{\text{dmp}} \tag{1}$$

The first term $\tau_{\text{trans}}$ is the elastic stress transfer resulted from displacements over the entire fault. The second term $\tau_{\text{per}}$ is the perturbation term from external stress loading (not considered in this study). The third term $\tau_{\text{dmp}}$ is the so-called radiation damping that partially accounts for the dynamic effect via radiation energy of shear waves[55], see more in[56]. The fault friction $\mu$ is a function of slip rate $V$ and state $\theta$, controlled by constitutive parameters of direct and indirect effect $a, b$

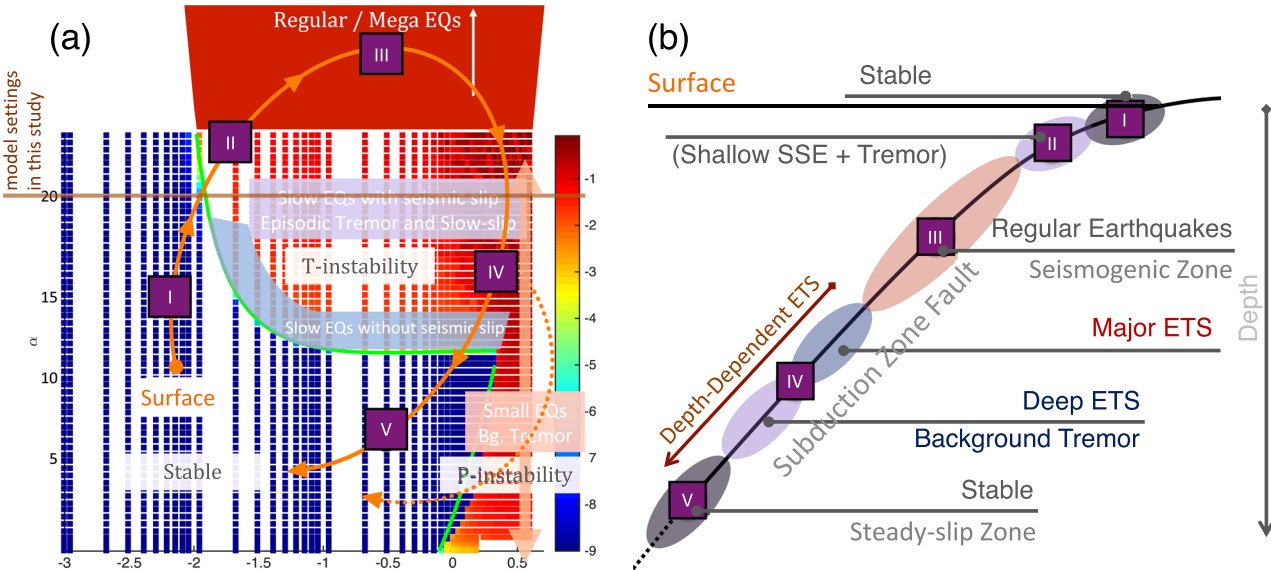

**Fig. 9 A concept model of unified mechanism of slow-to-fast earthquakes for subduction zone fault. a** Fault stability diagram. Color shows the logarithmic max slip rate (log10 of m · s⁻¹) of heterogeneous faults with various ratio of asperity/matrix effective normal stress $\alpha = \sigma_{\text{asp}}/\sigma_{bg}$ (asperity effective normal stress/background matrix effective normal stress, y-axis) and asperity criticalness $\beta = L_{\text{asp}}/L_c$ (asperity size/critical length, x-axis). Red: seismic, blue: steady-slip, turquoise-yellow: slow-slip. With variation of $\alpha$ and $\beta$ fault can transit from stable (steady-slip) to slow and fast earthquakes (path I–V). **b** Schematic view of generalized subduction zone fault with depth-dependent episodic tremor and slow-slip.

and characteristic slip distance $D_c$:

$$\mu = \mu^* - a\ln\frac{V^*}{V} + b\ln\frac{V^*\theta}{D_c} \quad (2)$$

where $\mu^*$ is the reference friction coefficient, $V^*$ is the reference slip rate. The state variable $\theta$ evolves with time and is described by laboratory-derived empirical evolution laws, the so-called aging-law and slip-law are the two types of evolution laws commonly used[54]. In this study we adopt the "slip-law" that is more consistent with laboratory experiments[57]:

$$\overset{*}{\theta} = -\frac{V\theta}{D_c}\ln\frac{V\theta}{D_c} \quad (3)$$

The constitutive parameters $a, b$ competitively determine material's property that affects the fault's stability. When $a - b > 0$, the material is VS as it strengthens (increase of friction) with an increase in slip rate, serving as a negative feedback to the slip rate. Thus, fault is stable (no spontaneous events, steady-slip at loading rate). When $a - b < 0$, the material is VW as it weakens (decrease of friction) with an increase in slip rate, serving as a positive feedback to the slip rate. Thus, fault is unstable (generating spontaneous events) if the fault is "supercritical," i.e., the size of fault exceeds certain critical size $L_c$. The critical size $L_c$ is a direct measurement of fault criticalness determined by its frictional properties and effective normal stress. For a homogeneous fault, it is simply as:

$$L_c = C_g\frac{GD_c}{\sigma(b-a)} \quad (4)$$

$C_g$ is a constant determined by fault geometry, for simplicity it is treated as 1 in this study.

Opposite to the supercritical case, if the fault size is smaller than $L_c$, it is "subcritical" and no spontaneous event will occur albeit the fault being VW. In this study we explored asperities with a range of individual criticalness $\beta = L_{asp}/L_c$. The criticalness of the fault with heterogeneous frictional properties and stress condition, like the AIM model considered in this study, is more complicated and usually need to resort to homogenization or numerical solution. Luo and Ampuero[25] conducted a complete analysis of fault criticalness of a heterogeneous fault, see also[58–61] and reference therein for more details of fault stability and homogenization of heterogeneous faults.

**The asperity-in-matrix fault model**. Our heterogeneous AIM fault model consists of alternating VS–VW materials (VS fault matrix with embedded VW tremor asperities) and differential effective normal stress[38,62,63]. This geologically inspired subduction fault model is physically sound[33,34,63–65] and based on previous modeling efforts of composite fault with mixed VW–VS rheology[25,60,61,66]. In this model the competent embedment of the subduction zone material is interpreted as VW tremor asperity as it can rupture with high slip rate and emitting seismic NVT signals, while the incompetent background is interpreted as VS matrix so it dampens the post-seismic rupture of tremor asperities and remains aseismic (slow-slip). The interplay of VW and VS materials jointly affects ETS behaviors with slow (SSE) to fast (NVT) signature.

In this study we develop the AIM models with depth-dependent stress variations utilizing the open-sourced rate-and-state earthquake simulator QDYN[56] (see last subsection for details). As shown in Fig. 3, we consider a 2-D fault model in 3-D medium with only the seismic-aseismic transition zone from 30 to 50 km, slightly smaller than the depth range of ETS zone in CSZ to reduce computational cost while still being able to capture the essence of various ETS patterns. The boundary used to characterize ETS distribution for the "mid" and "deep" depth is instead 44 km (Figs. 5, 6, and 8b) as compared to 46 km in observation (Figs. 1, 2, and 8a), to accommodate the reduction of ETS zone depth. We use a planar fault with a constant dipping angle of 20° resulting in total dip-parallel fault width about 58.5 km. The strike-parallel fault length is 128 km, representing a subset of northern ETS segment of CSZ and is sufficient to reproduce steadily propagating major ETS events. The fault is discretized with 500 m by 500 m (length by width). To further reduce the computational cost the tremor asperity is prescribed in the ETS zone as single-cell VW patch ($L_{asp} = 500m$) so we do not resolve the rupture inside the tremor asperity. Using single-cell tremor patch of relatively large size reduces computational expense significantly, allowing us to study full 3-D effects with multiple ETS cycles over a longer time scale. However, rupture details within each tremor asperity are not resolved, thus detailed source statistics such as source time function, spatial-temporal distribution of tremor (or LFE) magnitudes cannot be fully recovered. Nevertheless, since we focus on the macroscopic behavior of the ETS, we consider that the lack of detailed rupture process inside individual tremor asperity will not change major conclusion we draw from this study. The effective normal stress ($\sigma$) of the tremor asperity is 20 times the value of surrounding VS background ($\alpha$) making it break more seismically and generating ETS (Fig. 9, see also[25,38]). All the asperities have the same constitutive parameters ($\alpha$ and b) but the characteristic slip distance $D_c$ value is randomly chosen such that each tremor asperity has a random criticalness ($\beta = L_{asp}/L_c$, Fig. 9, Luo and Ampuero[25]) ranging from 0.2 to 1.0 (Fig. 4b and Supplementary Figs. 3a and 4a). Note that the tremor asperity has a higher criticalness compared to previous study[38] in order to generate "background tremor." Tremor asperities are placed in the ETS region with

a hexagon style pattern (Fig. 3) such that each tremor asperity has nearly the same distance to its six neighboring asperities with a VW/VS area ratio of 1/11 in the ETS region. Two separate tests with tremor asperities arranged in either rectangular pattern or randomly distributed pattern yield similar macroscopic results (Supplementary Fig. 6). Thus, we conclude that the ratio of VW/VS rather than the actually spatial patterns of tremor asperity distribution controls the overall behavior of the VW/VS mixed fault. The choice of the VW/VS ratio and cell size is a trade-off between observational[34,35], insights from analytical studies[25], and limitation of our current computational capabilities. Constant loading of $V_{pl} = 10^{-9}\text{m} \cdot \text{s}^{-1}$, representing tectonic loading rate of 31.5 mm per year, is applied across the fault.

To test our hypothesis that depth-dependent stress condition is the dominant control factor of depth-dependent ETS patterns, we consider three AIM scenarios with different effective normal stress profiles. The first is a reference model of a single ETS zone with uniform effective normal stress (1 MPa for the VS background and 20 MPa for the VW tremor asperities, respectively), thereafter referred as "uniform model." The second is a model with two connected ETS zones, the shallower ETS region (above 40 km depth) has the same setting as the uniform model, whereas the deeper ETS region (below 40 km depth) has 1/10 effective normal stress of the shallower counterpart (0.1 MPa background, 2 MPa asperities) so that the deeper ETS region is expected to have weaker but more frequent ETS events, thereafter referred as "bi-modular model." The third model considers effective normal stress of the VS background and VW tremor asperities as a linear function of depth: from 30 to 50 km of 1–0.1 MPa for VS background and 20–2 MPa for the VW tremor asperities. We refer this model as "linear model." Details of model settings can be referred to Fig. 3, Supplementary Table 1, and Supplementary Figs. 3 and 4. The parametric settings loosely follow our previous study[25,63] as appropriate for the ETS zone of CSZ with observational constraints. Nevertheless, various frictional properties, including constitutive parameters $\alpha$ and b, will also affect the stability of the fault[61,66] and was not fully explored in this study. In our model the effects of the frictional parameters $\alpha$ and b can trade-off with effective normal stress ($\sigma$) as the ratio of $a\sigma/b\sigma$ controls the fault stability (here $\sigma$ will not cancel out when fault is heterogeneous). Even though another set of frictional parameters is possible, we choose not to do an exhaustive search as that is not the main focus of the study. Instead, we focus on using the $\alpha$ and b values derived from our previous study that are constrained by available observations to model ETS behaviors[38].

Rate-and-state numerical simulations of the three ETS models are performed with QDYN for 4 years each so that multiple occurrences of major ETS can be captured. We then analyze the ETS patterns produced by each model discarding the first year of output ("warm-up" period) to mitigate the effects of the initial conditions. We detect tremor using a simple velocity threshold $V_{th}$ (equal to $10^4 V_{pl}$) by examining the local slip rate at tremor asperity. This definition is different from the conventional seismological definition of tremor, of which continuous tremor is characterized as a train of weak seismic signals predominantly at frequencies of 1–8 Hz, with very similar waveforms and spectra as a collection of LFE (Ide et al. 2007; Shelly et al., 2007). Given the small size of the source we considered a general view about tremor generation as collection of individual LFEs[17,36,37], we consider it a good first-order approximation. Various $V_{th}$ (equal to $10^3$ and $10^5 V_{pl}$) have been examined (Supplementary Fig. 7) and we find varying the detecting threshold mostly uniformly affects the tremor counts within each ETS thus will not affect any main results of this paper.

**Estimating the magnitude of ETS**. The magnitudes for ETS and tremor (LFE) in the real world are important yet challenging to determine[22,43]. Here we conveniently use total moment $M_0$ to estimate the magnitude of our tremor and ETS, defined as[67]

$$M_0 = GAD \quad (5)$$

Where $G$ is shear modulus, $A$ is the rupture area, and $D$ is the average slip in the rupture area. Here we do not distinguish seismic and aseismic slip so $M_0$ represents the total moment including both seismic and aseismic moment.

We then estimate the moment magnitude ($M_w$) of tremor and ETS as[68]

$$M_w = \frac{2}{3}\log_{10}(M_0) - 10.7 \quad (6)$$

In this study, for single tremor events (LFE), including stand-alone tremor and/or individual tremor within an ETS event, $A = 500m \cdot 500m$ (single cell), and estimated average slip ($D$) per tremor event from our rate-and-state modeling results is in the order of ~1 mm to 1 cm, these give tremor $M_w$ estimates of ~2.5–3.2. On the other hand, simple extrapolation for tremor asperities of size ~100 m results in $M_w$ ~ 1.5–2.3 with fixed $D$, or $M_w$ ~ 0.7–1.4 if tremors are self-similar ($D$ is proportional to rupture length). The latter is more consistent with the observations[43,69]. For ETS events, we estimated $M_w$ of major ETS in our model to be ~6.0–6.4 with ~1–3 cm of average slip, bounded by relatively small size of the modeled fault segments (128 km in length). If we allow the fault to be 500 km in length then ETS $M_w$ can reach 6.8. Meanwhile, deep ETS has average slip in the order of millimeters resulting in $M_w$ ~ 4.5–6.0.

**Quasi-dynamic earthquake simulator QDYN.** We utilize software QDYN[56] to perform all rate-and-state numerical simulations of ETS. QDYN is a boundary element software to simulate earthquake cycles (seismic and aseismic slip on tectonic faults) under the quasi-dynamic approximation (quasi-static elasticity combined with radiation damping) on faults governed by rate-and-state friction and embedded in elastic media. QDYN includes various forms of rate-and-state friction and state evolution laws, and handles non-planar fault geometry in 3-D and 2-D media, as well as spring-block simulations. Loading is controlled by remote displacement, steady creep or oscillatory load. In 3-D it handles free surface effects in a half-space, including normal stress coupling. The medium surrounding the fault is linear, isotropic, and elastic, and may be uniform or (in 2-D) contain a damaged layer. QDYN implements adaptive time stepping, shared, and distribute memory parallelization (OpenMP/MPI), and can deal with multi-scale earthquake cycle simulations with fine details in both time and space. It is equipped with user-friendly MATLAB and Python interfaces and graphical output utilities. QDYN is published[56] and available at https://github.com/ydluo/qdyn. More details and instruction can be found there.

## Data availability

The tremor catalog used in this study is publicly available at the PNSN website (https://pnsn.org). All rate-and-state simulation results from this study are published via Dataverse at https://doi.org/10.25346/S6/QSIOG1[72]. No other data are used in this study.

## Code availability

The rate-and-state earthquake simulator (QDYN) used in this study is open-sourced and publicly available at https://github.com/ydluo/qdyn.

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

## Acknowledgements

Part of the concept from this study originates from the idea and discussion Y. L. had during his thesis study with his PhD advisor Prof. Jean-Paul Ampuero. Y.L. thanks J.-P. Ampuero for the intellectual contribution. This work was carried out at the Jet Propulsion Laboratory, California Institute of Technology, under a contract with the National Aeronautics and Space Administration.

## Author contributions

Y.L. came up the idea of stress variation on ETS. Z.L. helped refine the idea. Y.L. conducted the analysis. Z.L. supervised the project. Both Y.L. and Z.L. contributed to writing the manuscript.

## Competing interests

The authors declare no competing interests.
