## [Peer Review File · Nature Communications]

REVIEWER COMMENTS

Reviewer #1 (Remarks to the Author):

Review of "Fault Zone Heterogeneities Explain Depth-dependent Pattern and Evolution of Slow Earthquakes in Cascadia"

In this study, the authors use a numerical model with heterogeneous asperities-in-matrix (AIM) fault to study the tremor and slow-slip events (ETS) in the Cascadia Subduction Zone (CSZ). They incorporate along-dip variations of the differential pore pressure to the model and consider three end-member scenarios: a reference model with no depth-dependent stress variation (uniform), a second one with a sharp transition of effective normal stress with depth (bi-modular), and a third one with gradual decrease in depth-dependent effective normal stress (linear). They find that the linear model is capable of quantitatively reproducing similar observations as seen in the region, i.e. behaviors that range from small individual tremor, to arrested and runaway deep ETS, to the characteristic major ETS.

Advancing our understanding on the full spectrum of earthquake fault processes and developing physics-based fault models capable of explaining that is crucial for seismic hazard assessments, especially in high-risk regions such as the CSZ. This is an original study that reproduces the full complexity of ETS patterns and the depth-frequency observed in the CSZ and should be of interest to readers in different fields. I would also be highly interested to see a future study with the AIM model that incorporates both the megathrust earthquake region and the ETS region, as mentioned by the authors.

The numerical tool used in this analysis is well designed and the conclusion of the manuscript is well supported by the analyses. The manuscript is also nicely written and organized. The authors include a thorough background of the observations in the transition zone of the CSZ, as well as the development of various fault models thus far and their limitations, which suits well for journals like Nature Communications with a broad audience. Figures are also nicely designed and clearly explained. I think it would merit the publication in Nature Communications. Below are my comments which I hope the authors can address in their revision:

- Can the authors provide the basis for prescribing a VW/VS ratio of 1/11 and $L_{\text{asp}} = 500$ m? Are these comparable to field observations of exhumed subduction fault zone materials?
- The authors state in the manuscript that setting V_{th} as $10000 V_{\text{pl}}$ is a first-order approximation, but since that is a major variable in defining tremors in the simulations, it would be helpful if the authors can comment on the variance of their numerical results if they assign different values of V_{th} .
- In the rate-and-state framework, the properties of slow slip on a fault are heavily controlled by the frictional parameters a and b . The numerical model presented by the authors is prescribed with only one set of frictional parameters. Though I understand that this is not the main focus of this paper, it would strengthen the analysis overall if the authors can also discuss how their results may deviate from the those presented in this study if they model with different (yet reasonable) sets of a and b , and whether their model can shed lights on the inference of the frictional properties of the study area.

Minor comments:

- Line 77 and 80: I think citations are needed for the two sentences "Even though the associated slow-slip..." and "In fact, the major, deep and arrested ETS are very similar..."
- L258: with "those" instead of "these"?
- I think the first sentence in the second paragraph on page 9 (lines 270-271) is unnecessary. It describes the key result of this manuscript, which has already been stated in the article for a few times, but doesn't really connect with the key ideas of this paragraph.

Reviewer #2 (Remarks to the Author):

In this paper, Luo and Liu build a subduction zone fault model within a 3D elastic medium. The fault includes a distribution of velocity-weakening asperities embedded within an otherwise velocity-strengthening matrix, motivated by geological and geophysical observations of slow fault slip. A uniform loading rate is applied to the model and the authors monitor the changes in fault slip as it progresses during a cycle of several years. Slow wave fronts are observed and are classified as Episodic Tremor and Slip (ETS) events. Tremor is defined as a slip velocity threshold of 10,000 times the background loading rate. The authors explore three models where properties of the asperities and matrix (effective stress and critical slip distance) change as a function of depth, in order to examine the effects on the macroscopic evolution of fault slip. The first model retains uniform properties across the entire fault. The second model imposes an abrupt discontinuity in effective stress along dip. The third model proposes a linear change in effective stress along dip. The latter model leads to a wide range of fault behaviors that are also observed in nature, including: 1) segmentation of ETS into shallow (large magnitude) and deep (lower magnitude) slow slip; 2) general initiation of ETS in the deeper segment; 3) Rapid tremor reversals; 4) ETS migration along strike; 5) bi-directional ETS along-strike; 6) deep arrested ETS; and 7) change in ETS recurrence as a function of depth (along-dip distance).

Taken together, this model does a truly amazing job at reproducing most macroscopic observations of slow slip and tremor propagation, at least in Cascadia. The model is based on geological observations that describe the asperity/matrix embedding, and geophysical observations that suggest increasing pore-fluid pressure in the slow slip source region. The paper is well written, and I recommend publication after the authors address a few comments below.

1 - Definition of tremor:

The authors define a single tremor event as an asperity with slip rate greater than 10,000 times the background loading rate. This seems rather arbitrary, and it isn't clear that slip on such an asperity would reproduce the typical properties of a single tremor (or low-frequency) event: 1) constant duration – magnitude scales with slip; or 2) repetition (each LFE/tremor source in space can generate hundreds of events during each ETS). Can this be observed/quantified?

2 - Magnitudes:

The authors state that "the linear stress model is able to reproduce a much richer set of ETS events with various magnitudes and depth" (lines 160-161; Figure 5), although magnitude is never properly determined. Can the authors measure macroscopic slow slip magnitude and compare with observations in Cascadia? This will likely depend on the specific model geometry (e.g., size of 2D fault, since any given shallow ETS seems to rupture the entire fault), but it would be good to provide some quantification of this. Furthermore, the range of LFE magnitude is typically $M_w \sim 1-2.5$ (Bostock et al., 2015). Is this reflected in the tremor distribution?

3 - Distribution of asperities:

How important is the exact distribution of asperities in the model? Is a geometrical pattern (e.g., hexagonal, rectangular) necessary to obtain slow slip fronts? Is it possible to use a random distribution of asperities that maintains the 1/11 asperity/matrix ratio? How sensitive are these results to different ratios?

4 - Other macroscopic behavior:

The observed spatio-temporal distribution of LFE magnitudes indicates that half the moment release occurs within the first 12-24 hours of LFE activity, during an ETS episode; The remainder is released in bursts over several days, particularly as RTR during which tidal sensitivity is high (Bostock et al., 2015). Is there a way to quantify this effect in the models?

Furthermore, is there a way to perturb the model to simulate transient stress due to external sources (e.g., surface waves) and observe dynamic triggering or tremor (e.g., Rubinstein et al., 2009)? If the fault model is sensitive to the stress level as mentioned, this could be another relevant observation in favor of the model (e.g., when in the ETS cycle is tremor triggered by stress transients, etc.)

Does the model reproduce secondary slip fronts following the main front? This would be observed as dynamic bursts of tremor activity and slow slip continuing in the wake of the main front (e.g., Peng et al., 2015; Bletery et al., 2017).

If the model(s) do reproduce these features, this would further highlight the strength of the model. If not, it would be important to discuss the reasons why these features/phenomena are not or cannot be observed (not enough physics included? Limitations due to the modeling strategy? Etc.). Essentially, what are the main micro- to macroscopic phenomena that the model does not/cannot reproduce, other than large slip in the seismogenic portion?

References:

M G Bostock, A M Thomas, G Savard, L Chuang, and A M Rubin. Magnitudes and moment-duration scaling of low-frequency earthquakes beneath southern Vancouver Island. *Journal of Geophysical Research: Solid Earth*, 120(9):6329–6350, 2015.

Justin L Rubinstein, Joan Gomberg, John E Vidale, Aaron G Wech, Honn Kao, Kenneth C Creager, and Garry Rogers. Seismic wave triggering of nonvolcanic tremor, episodic tremor and slip, and earthquakes on Vancouver Island. *Journal of Geophysical Research*, 114, 2009.

Yajun Peng, Allan M Rubin, Michael G Bostock, and John G Armbruster. High-resolution imaging of rapid tremor migrations beneath southern Vancouver Island using cross-station cross correlations. *Journal of Geophysical Research: Solid Earth*, 120(6):4317–4332, 2015.

Quentin Bletery, Amanda M Thomas, Jessica C Hawthorne, Robert M Skarbak, Alan W Rempel, and Randy D Krogstad. Characteristics of secondary slip fronts associated with slow earthquakes in Cascadia. *Earth and Planetary Science Letters*, 463:212–220, 2017.

Reviewer #3 (Remarks to the Author):

Some comments.

1. Please consider same/blue color for the outline and filling of the blue-triangles in the figures. Likewise, for the red-triangles. Overall, the figures have lots of scope for improvement. I guess that full-width may be appropriate for the plots in the Figure 5d,e,f. Likewise, in Figure 8a,b. Figure 1a is aesthetically appealing.

Figure 9: The units of the slip rate is misisng. In the caption the terms 'x-axis' and 'y-axis' are interchanged.

*Color choices for plots $D < c >$ (Figures 4, S3, S4) needs improvements. Please choose a different color scheme with more contrast. It may be lot better to consider a surface plot for the variation of $D < c >$.

2. Apart from the intrinsic scale $L < c >$, the model introduces few lengthscales through: (i) The variations of the D_c along the strike and the dip. (ii) Size of the asperities and their spacings along the dip and the strike. (iii) a non-uniform fault normal stress. These scales, together, influence the results: migration, reversals and propagation. Taken together, the fault heterogeneity considered in this study is not simple. A short discussion about their relative importance may be desired. Please consider showing the distribution of the criticalness parameter, β , in Figures 4, S3, S4. Preferably (aesthetically appealing) surface plots and one dimensional along dip and strike variations.

3. Are all parameter variations simultaneously important? Is there any homogenization scale as suggested in Ray and Viesca (GJI, 2019) on homogenization of fault frictional properties?

4. It may be good to highlight how asperity-in-matrix model is similar to or different from the heterogeneities considered in a PhD student's thesis with title "Earthquake nucleation with heterogeneous physical properties". Therein the chapters sort of touched some aspects of heterogeneities and has some results related to slip migration along the fault to long distances.

5. What parameters are considered to mean VW/VS ratio (line-379)?

6. The method section has lots of scope for improvements in terms of tight presentation. Many papers, mainly standard theoretical ones, are not cited. For example, consider citing Rice (1993) where radiation damping is introduced.

7. Consider following corrections:

Correct for symbol inconsistency: $D_{<c>}$ or D_c

Line 51-52: "CSZ displays clear along-strike variation.....". Please remind along-strike variation of what?

Line 137: typo corrected: heterogeneous.

Line: 273: please write all that is meant by 'etc.'

Line 275-278: Please remind the readers about the scaling relations.

Line 403-406: Please remind the readers the conventional seismological definition of tremor briefly. To what quantity the phrase 'a good first order approximation' referring to (also Line 72)? It may be suitable just to write slip rate of $1000Vpl$ is an indicator of tremor type situations.

We sincerely thank all the reviewers and editor for their favorable consideration and valuable comments for the manuscript. All comments have been addressed and/or clarified and the manuscript has been updated accordingly. Kindly please find the authors' point-to-point response below in BLUE text.

REVIEWER COMMENTS

Reviewer #1 (Remarks to the Author):

Review of "Fault Zone Heterogeneities Explain Depth-dependent Pattern and Evolution of Slow Earthquakes in Cascadia"

In this study, the authors use a numerical model with heterogeneous asperities-in-matrix (AIM) fault to study the tremor and slow-slip events (ETS) in the Cascadia Subduction Zone (CSZ). They incorporate along-dip variations of the differential pore pressure to the model and consider three end-member scenarios: a reference model with no depth-dependent stress variation (uniform), a second one with a sharp transition of effective normal stress with depth (bi-modular), and a third one with gradual decrease in depth-dependent effective normal stress (linear). They find that the linear model is capable of quantitatively reproducing similar observations as seen in the region, i.e. behaviors that range from small individual tremor, to arrested and runaway deep ETS, to the characteristic major ETS.

Advancing our understanding on the full spectrum of earthquake fault processes and developing physics-based fault models capable of explaining that is crucial for seismic hazard assessments, especially in high-risk regions such as the CSZ. This is an original study that reproduces the full complexity of ETS patterns and the depth-frequency observed in the CSZ and should be of interest to readers in different fields. I would also be highly interested to see a future study with the AIM model that incorporates both the megathrust earthquake region and the ETS region, as mentioned by the authors.

The numerical tool used in this analysis is well designed and the conclusion of the manuscript is well supported by the analyses. The manuscript is also nicely written and organized. The authors include a thorough background of the observations in the transition zone of the CSZ, as well as the development of various fault models thus far and their limitations, which suits well for journals like Nature Communications with a broad audience. Figures are also nicely designed and clearly explained. I think it would merit the publication in Nature Communications. Below are my comments which I hope the authors can address in their revision:

We are grateful to Reviewer #1 for the constructive comments and positive evaluation of our manuscript. We are currently developing AIM models incorporating both the megathrust earthquake region and the ETS region and plan to publish results in a future study. We have carefully considered all the comments and below please find our respective answers to individual points.

1 - Can the authors provide the basis for prescribing a VW/VS ratio of 1/11 and $L_{\text{asp}} = 500$ m? Are these comparable to field observations of exhumed subduction fault zone materials?

We consider the VW/VS ratio and asperities size based on several constraints: (1) Geological observation of exhumed fault with a mix of competent asperities and incompetent background materials (analogous to VW and VS materials, e.g., Fagereng, 2011; Fagereng and Cooper, 2010; Fagereng and Sibson, 2010). Note that the actual competent asperity size may or may not represent the tremor source size as the tremor can be a collective failure of multiple competent asperities. (2) Seismology and geodesy results of tremor and ETS showing tremor moment release is only a very small fraction (<1%) of total moment release, suggesting a sub-100 or 100 meters radius for tremor source (Kao et al., 2010). (3) Results from previous analytical study (Luo and Ampuero, 2018), which did a complete study of heterogeneous fault with VS-VW mix and pore-pressure difference with various VW/VS ratio. The study shows the actual size of VW patch (L_{asp}) won't affect the overall pattern of fault stability but do indeed affect the actual size (magnitude) of individual tremor events (See also Yabe and Ide 2017, 2018). Thus, the actual choice of the VW/VS ratio and cell size is a balance between observational constraints, insights from analytical studies (also refer to our response to reviewer 3 comment #3), and our current computational limitations. Kindly refer to our reply to question #2 and #3 for review 2 with related discussions on VW/VS ratio and sizes of tremor asperities.

We have updated our manuscript accordingly to justify our choice of model settings, including but not limited to VW/VS ratio and L_{asp} . (Line 403-407, Line 429-433 of tracked change version and Line 381-385, Line 398-402 of clean version).

2 - The authors state in the manuscript that setting V_{th} as $10000 V_{\text{pl}}$ is a first-order approximation, but since that is a major variable in defining tremors in the simulations, it would be helpful if the authors can comment on the variance of their numerical results if they assign different values of V_{th} .

We agree that V_{th} is a key variable for defining tremors in our simulations. We have tried different V_{th} and the overall ETS pattern remains the same even when we change V_{th} by a factor of 100 (see Figure R1). Naturally a lower V_{th} results in more tremor events to be

detected, which as a first-order approximation mostly just make a (nearly) constant multiplication of tremor counts per ETS, as shown in the plot. Thus we conclude that variation of V_{th} value won't affect any main conclusion of this manuscript. We have included the discussion of tremor detection threshold in the updated manuscript. (Line 468-471 of tracked change version and Line 435-438 of clean version).

Figure R1: Comparison of different tremor detecting threshold V_{th} . (a-c): Similar to Figure 5 in main text, showing detected tremor activities with strike- and dip-parallel projection, and tremor activities binned in 5-day window. (a) $V_{th} = 100000 V_{pl}$; (b) $V_{th} = 10000 V_{pl}$; (c) $V_{th} = 1000 V_{pl}$; The overall ETS patterns remain the same despite V_{th} is varied by a factor of 100. (d) Global maximum slip rate as a function of time with three detecting thresholds used in (a-c) as solid red, green and blue lines.

3 - In the rate-and-state framework, the properties of slow slip on a fault are heavily controlled by the frictional parameters a and b . The numerical model presented by the authors is prescribed with only one set of frictional parameters. Though I understand that this is not the main focus of this paper, it would strengthen the analysis overall if the authors can also discuss how their results may deviate from the those presented in this study if they model with different (yet reasonable) sets of a and b , and whether their model can shed lights on the inference of the frictional properties of the study area.

The effects of parameters a and b of a fictionally heterogeneous fault have been extensively studied using analytical and numerical methods in previous studies (e.g., Skarbek et al., 2012; Yabe and Ide, 2017, 2018). Similarly, the effects of other frictional parameters (e.g., D_c) and stress heterogeneities have been addressed in Luo and Ampuero (2018). In our model the effects of the frictional parameters a and b can trade-

off with effective normal stress (σ) as the ratio of $a\sigma/b\sigma$ controls the fault stability (here σ will not cancel out when fault is heterogeneous). Even though other set of frictional parameters is possible, we choose not to do an exhaustive search as that is not the main focus of the study. Instead, we focus on using the a , b values derived from our previous study that are constrained by available observations to model ETS behaviors (Luo and Liu, 2019b). Nevertheless, we add some discussion to acknowledge the effects of other possible frictional parameters that might contribute to the complexity of model results. (Line 448-455 of tracked change version and Line 417-424 of clean version).

.

Minor comments:

- Line 77 and 80: I think citations are needed for the two sentences “Even though the associated slow-slip...” and “In fact, the major, deep and arrested ETS are very similar...”

We now add references of Michel et al. We now explicitly mentioned in the manuscript we find that the major, deep and arrested ETS are very similar in the early phase. (Line 79-83 of tracked change version and Line 79-83 of clean version).

- L258: with “those” instead of “these”?

It's fixed.

- I think the first sentence in the second paragraph on page 9 (lines 270-271) is unnecessary. It describes the key result of this manuscript, which has already been stated in the article for a few times, but doesn't really connect with the key ideas of this paragraph.

The sentence was removed. The paragraph now starts with “Our AIM-based model also reveals some good similarities between hierarchical ETS patterns and ordinary earthquakes...”. (Line 284 of tracked change version and Line 278 of clean version).

Reviewer #2 (Remarks to the Author):

In this paper, Luo and Liu build a subduction zone fault model within a 3D elastic medium. The fault includes a distribution of velocity-weakening asperities embedded within an otherwise velocity-strengthening matrix, motivated by geological and geophysical observations of slow fault slip. A uniform loading rate is applied to the

model and the authors monitor the changes in fault slip as it progresses during a cycle of several years. Slow wave fronts are observed and are classified as Episodic Tremor and Slip (ETS) events. Tremor is defined as a slip velocity threshold of 10,000 times the background loading rate. The authors explore three models where properties of the asperities and matrix (effective stress and critical slip distance) change as a function of depth, in order to examine the effects on the macroscopic evolution of fault slip. The first model retains uniform properties across the entire fault. The second model imposes an abrupt discontinuity in effective stress along dip. The third model proposes a linear change in effective stress along dip. The latter model leads to a wide range of fault behaviors that are also observed in nature, including: 1) segmentation of ETS into shallow (large magnitude) and deep (lower magnitude) slow slip; 2) general initiation of ETS in the deeper segment; 3) Rapid tremor reversals; 4) ETS migration along strike; 5) bi-directional ETS along-strike; 6) deep arrested ETS; and 7) change in ETS recurrence as a function of depth (along-dip distance).

Taken together, this model does a truly amazing job at reproducing most macroscopic observations of slow slip and tremor propagation, at least in Cascadia. The model is based on geological observations that describe the asperity/matrix embedding, and geophysical observations that suggest increasing pore-fluid pressure in the slow slip source region. The paper is well written, and I recommend publication after the authors address a few comments below.

We are truly thankful to Reviewer #2 for the supportive comments and valuable feedback for our manuscript. We have carefully considered all the comments. Below please find our response to various points raised.

1 - Definition of tremor:

The authors define a single tremor event as an asperity with slip rate greater than 10,000 times the background loading rate. This seems rather arbitrary, and it isn't clear that slip on such an asperity would reproduce the typical properties of a single tremor (or low-frequency) event: 1) constant duration – magnitude scales with slip; or 2) repetition (each LFE/tremor source in space can generate hundreds of events during each ETS). Can this be observed/quantified?

Tremor is usually considered as a collection of LFEs ("individual tremor" as in this manuscript), thus we conveniently use a slip-rate threshold detection mechanism for each individual tremor patch. With this point source model, the constant duration / magnitude vs slip is natural.

The reactivation of the LFE/tremor source is indeed reproduced by our model, and in fact, we think that's exactly the origins of Rapid Tremor Reversals (and tremor streaks). We have updated the manuscript to better emphasize that point. (Line 161, 270-271, 276-278 of tracked change version and Line 158, 264-265, 270-272 of clean version).

In our model tremor source does not reactivate (or repeat) hundreds of times during each ETS. One reason is that due to computational limitations we can only consider a rather sizable tremor source (500m * 500m) thus not be able to resolve the slip details of each tremor. This compromise enables us to study the macroscopic behavior of ETS over multiple ETS cycles but at the cost of missing detailed rupture processes within the tremor patch. We have added discussion in the updated manuscript. (Line 403-407 of tracked change version and Line 381-385 of clean version).

For the selection of tremor threshold (10000 Vpl), please see our response to comment 2 of review #1 for more details

2 - Magnitudes:

The authors state that "the linear stress model is able to reproduce a much richer set of ETS events with various magnitudes and depth" (lines 160-161; Figure 5), although magnitude is never properly determined. Can the authors measure macroscopic slow slip magnitude and compare with observations in Cascadia? This will likely depend on the specific model geometry (e.g., size of 2D fault, since any given shallow ETS seems to rupture the entire fault), but it would be good to provide some quantification of this. Furthermore, the range of LFE magnitude is typically $M_w \sim 1-2.5$ (Bostock et al., 2015). Is this reflected in the tremor distribution?

We could use total moment $M_0 = GDA$ to get a first order estimation of the magnitude of simulated tremor events, which include both seismic and aseismic moments released during an event (thus likely an over-estimation compared with real-world observation). In our model shear modulus G is 30GPa. For tremor asperities the area A is 500m*500m (single cell), and measured slip (D) per tremor event is in the order of ~1 mm to 1cm. These give tremor M_w estimates of ~2.5-3.2, which is on the higher end of Bostock et al., (2015). On the other hand if we assume ~100m*100m tremor asperities, estimated M_w will be ~ 1.5-2.3 with fixed D , or $M_w \sim 0.7-1.4$ if tremors are self-similar (D is proportional to rupture length). In 2-D models (Luo and Liu 2019b) we used sub-hundred-meter tremor asperities that are in agreement with Kao et al., (2010). However, using such small cell size in 3-D incurs at least 40X more time complexity thus is computationally challenging. Instead, we find that using larger tremor asperities reproduces the rich macroscopic ETS patterns as observed in Cascadia.

For major ETS our estimated Mw is ~ 6.0-6.4 with an average slip of ~1-3 cm, bounded by relative small size of the modeled fault length (128 km) but consistent with SSE Mw (6.1-6.7) from geodetic measurements (Schmidt et al, 2010). If we allow fault segment to be 500km long then ETS Mw can reach 6.8, comparable to larger events reported (e.g., Kao et al., 2010; Schmalzle et al., 2014). In comparison deep ETS in our model has average slip in the order of millimeters and Mw ~4.5-5.5. We have updated the main text to include magnitude estimation. (Line 187-191, 216-217, 473-492 of tracked change version and Line 181-185, 210-211, 440-459 of clean version).

3 - Distribution of asperities:

How important is the exact distribution of asperities in the model? Is a geometrical pattern (e.g., hexagonal, rectangular) necessary to obtain slow slip fronts? Is it possible to use a random distribution of asperities that maintains the 1/11 asperity/matrix ratio? How sensitive are these results to different ratios?

That's a good point. As we stated in the original manuscript "A separate test with tremor asperities arranged in a rectangular pattern yields similar macroscopic results so we conclude that the ratio of VW/VS rather than the actually spatial patterns of tremor asperity distribution controls the overall behavior of the VW/VS mixed fault."

Nevertheless, we have tested three different geometrical patterns of asperities:

(1) hexagonal (VW:VS = 1:11, same as used in main text of this study, resulting in 1581 VW asperities in the ETS region);

(2) rectangular (VW:VS = 1:11, VW asperities repeats every 5 cells along-strike and every 4 cells along-dip, resulting in 1581 VW asperities in the ETS region)

(3) random (each cell in the ETS zone has a 1/12 probability being a VW asperity, resulting in 1606 VW asperities in the ETS region). To simplify the problem, we change the range of sigma variation from 10 times to 2 times so only major ETS is reproduced. The results are summarized below in Figure R2. All the models are able to reproduce similar repeating major ETS with similar intervals (~0.9, ~0.9 and ~1.0 years, respectively).

In fact, Luo and Ampuero (2018) (e.g., Figure 3, also attached below for your convenience) have demonstrated that the VW/VS ratio ("f" in Luo and Ampuero, 2018) has an approximate linear relation with the inverse of "relative strength" ($\alpha = (|b-a|\sigma)_{vw} / (|b-a|\sigma)_{vs}$) which is the ratio between the amount of weakening in the VW area to the amount of strengthening in the VS area, due to heterogeneity of $|b-a|\sigma$.)

We have updated the manuscript (Line 417-429 of tracked change version and Line 395-398 of clean version) and added Figure S6 in supporting materials for the discussion of the exact distribution of asperities in the model.

Figure R2. Comparison of different asperity distributions. (a) hexagonal asperity distribution. (b) rectangular asperity distribution. (c) random asperity distribution. All the three models have same asperity criticalness (β) distribution (0.2-1.0) and VW/VS ratio (1/11). Left subplots: profile of asperity criticalness (β) and effective normal stress (σ). Right subplots: Global maximum slip rate as a function of time. The overall ETS patterns are very similar despite the actual spatial distribution of tremor asperities. Note to simplify the comparison the range of stress variation is 200%, instead of 1000% used in the Linear Model of main text, so only major ETS is reproduced.

Figure R3. Stability of the VW segment β , for VW/VS area ratios $f= 1/7$ (top) and $f= 1$ (bottom). Results of both QDYN simulations and linear stability analysis (LSA) are shown. Colored squares indicate the logarithm of peak slip rate (see color bar) reached in the VW segment in QDYN simulations, after the “warm-up” cycles. Each square is obtained from a separate simulation. Green solid curves are the stability boundaries determined by LSA (Linear Stability analysis). From Luo and Ampuero (2018).

4 - Other macroscopic behavior:

The observed spatio-temporal distribution of LFE magnitudes indicates that half the moment release occurs within the first 12-24 hours of LFE activity, during an ETS episode; The remainder is released in bursts over several days, particularly as RTR during which tidal sensitivity is high (Bostock et al., 2015). Is there a way to quantify this effect in the models?

This is related to your question 1, our model does reproduce LFE (tremor asperity) reactivation, but not quite close to the number of activities (hundreds of times) as in

Cascadia. We think this is mainly because we are modeling tremor collectively as a sizable single-cell asperity of 500m size. In order to get high spatial-temporal accuracy of LFE moment release information, we need to resolve rupture details and even might need to consider full dynamic effects. However due to computational limitations, resolving the rupture details for each individual LFE / tremor source is beyond the scope of this paper and our current modeling capability. We have updated our manuscript to be more explicit about current model limitations (Line 403-407 of tracked change version and Line 381-385 of clean version) and included Bostock et al., (2015) in our reference. (Line 488 of tracked change version and Line 455 of clean version).

Furthermore, is there is a way to perturb the model to simulate transient stress due to external sources (e.g., surface waves) and observe dynamic triggering or tremor (e.g., Rubinstein et al., 2009)? If the fault model is sensitive to the stress level as mentioned, this could be another relevant observation in favor of the model (e.g., when in the ETS cycle is tremor triggered by stress transients, etc.)

Yes, we are able to model external stress perturbation with our earthquake simulator QDYN. In Luo and Liu (2019b) we developed an approach to study both intrinsic variability of slow slip events and external perturbation effects from various sources including static co-seismic, dynamic co-seismic and post-seismic. Similar approach can be used to study the intrinsic variability of ETS and external stress loading effect. Nevertheless, a detailed study of ETS variability in response to stress perturbation from tectonic/non-tectonic sources is beyond the scope of this paper. We plan to address it in a future study.

Does the model reproduce secondary slip fronts following the main front? This would be observed as dynamic bursts of tremor activity and slow slip continuing in the wake of the main front (e.g., Peng et al., 2015; Bletery et al., 2017).

If the model(s) do reproduce these features, this would further highlight the strength of the model. If not, it would be important to discuss the reasons why these features/phenomena are not or cannot be observed (not enough physics included? Limitations due to the modeling strategy? Etc.). Essentially, what are the main micro- to macroscopic phenomena that the model does not/cannot reproduce, other than large slip in the seismogenic portion?

Yes, slip reactivation as RTRs and tremor streaks in the wake of the main front has been observed in our model. We have updated the manuscript to highlight and elaborate such observations. (Line 161, 270-271, 276-278 of tracked change version

and Line 158, 264-265, 270-272 of clean version). We also included the suggested references in the revised manuscript. (Line 277-278 of tracked change version and Line 271-272 of clean version).

References:

M G Bostock, A M Thomas, G Savard, L Chuang, and A M Rubin. Magnitudes and moment-duration scaling of low-frequency earthquakes beneath southern Vancouver Island. *Journal of Geophysical Research: Solid Earth*, 120(9):6329–6350, 2015.

Justin L Rubinstein, Joan Gomberg, John E Vidale, Aaron G Wech, Honn Kao, Kenneth C Creager, and Garry Rogers. Seismic wave triggering of nonvolcanic tremor, episodic tremor and slip, and earthquakes on Vancouver Island. *Journal of Geophysical Research*, 114, 2009.

Yajun Peng, Allan M Rubin, Michael G Bostock, and John G Armbruster. High-resolution imaging of rapid tremor migrations beneath southern Vancouver Island using cross-station cross correlations. *Journal of Geophysical Research: Solid Earth*, 120(6):4317–4332, 2015.

Quentin Bletery, Amanda M Thomas, Jessica C Hawthorne, Robert M Skarbek, Alan W Rempel, and Randy D Krogstad. Characteristics of secondary slip fronts associated with slow earthquakes in Cascadia. *Earth and Planetary Science Letters*, 463:212–220, 2017.

Reviewer #3 (Remarks to the Author):

Some comments.

1. Please consider same/blue color for the outline and filling of the blue-triangles in the figures. Likewise, for the red-triangles. Overall, the figures have lots of scope for improvement. I guess that full-width may be appropriate for the plots in the Figure 5d,e,f. Likewise, in Figure 8a,b. Figure 1a is aesthetically appealing.

Thank you for your suggestions, Figure 5 d,e,f and Figure 8 a,b are now in full width. We carefully compared triangular marks with and without outlines and found the ones with outline actually demonstrates better with high-resolution (print-quality)

representations. Attached is zoomed-in comparison for your reference. If you still think without outline is better, we will be happy to remove those.

Figure 9: The units of the slip rate is missing. In the caption the terms 'x-axis' and 'y-axis' are interchanged.

Figure 9 caption has been updated to include the unit of the slip rate. The x and y-axis mis-reference was fixed.

*Color choices for plots D_{c} (Figures 4, S3, S4) needs improvements. Please choose a different color scheme with more contrast. It may be lot better to consider a surface plot for the variation of D_{c} .

We tried several other color schemes but doesn't seem much improvement comparing with the original one, mainly because the range of D_c and contrast between the asperity and background. Surface plots also does not look good as tremor asperities are single cell. Figures are updated with asperities criticalness which we think should be a better representation of key model settings, also the criticalness is directly related (inversely proportional) to D_c .

2. Apart from the intrinsic scale L_{c} , the model introduces few lengthscales through: (i) The variations of the D_c along the strike and the dip. (ii) Size of the asperities and their spacings along the dip and the strike. (iii) a non-uniform fault normal stress. These scales, together, influence the results: migration, reversals and propagation. Taken together, the fault heterogeneity considered in this study is not simple. A short discussion about their relative importance may be desired. Please consider showing the distribution of the criticalness parameter, β , in Figures 4, S3, S4. Preferably (aesthetically appealing) surface plots and one dimensional along dip and strike variations.

That's a good point, various length scales do play a role in rate-and-state friction modeling, that's why non-dimensionalization is helpful whenever applicable (e.g., Luo and Ampuero 2018). However, we always need to consider the actual length scales when dealing with real-world problems. (i) L_c is proportional to D_c thus it's equivalent. (ii) Kindly refer to our reply to Question #1 of Reviewer #1, Question #3 of Reviewer #2 for size and distribution of tremor asperities. (iii) True, (ii) and (iii) jointly contribute to the asperity's individual, and collectively the fault's overall criticalness, and ultimately the modeled results. Please refer to our reply to your next question for more about parameter importance. We have updated our manuscript on the importance of fault heterogeneity in the model. (Line 365-370, 448-455 of tracked change version and Line 349-354, 417-424 of clean version). Figure 4, S3 and S4 are updated to show the criticalness parameter. We choose not to use surface plot as single-cell asperities do not look pretty (and sometime misleading) in surface plot.

3. Are all parameter variations simultaneously important? Is there any homogenization scale as suggested in Ray and Viesca (GJI, 2019) on homogenization of fault frictional properties?

It is tricky to test the relative importance of all frictional properties and stress conditions because of (1) trade-off effects, e.g., a , b trade-off with σ , kindly refer to our reply to Reviewer #1 Question #3. (2) The computational cost of problem with such large scale made it difficult to do a full parametric study thus we build 3-D models in this paper based on our previous results of ETS on a simpler 1-D fault model in Cascadia. Nevertheless, homogenization and uniformization of fault heterogenous frictional properties and stress conditions at various scales has been studied by a number of previous works, albeit with a much smaller scale model with simpler settings (e.g., Skarbak et al., 2012; Dublanchet et al., 2013 ; Yabe and Ide, 2017, 2018 and Luo and Ampuero, 2018). The instability boundary between total-instability (ETS event) and partial-instability (single tremor) can be approximately by homogenization equation (29) and (51) in Luo and Ampuero (2018). We now cite Ray and Viesca (2019) (Line 368-369 of tracked change version and Line 352-353 of clean version), and updated our manuscript accordingly. (Line 365-370, 448-455 of tracked change version and Line 349-354, 417-424 of clean version).

4. It may be good to highlight how asperity-in-matrix model is similar to or different from the heterogeneities considered in a PhD student's thesis with title "Earthquake nucleation with heterogeneous physical properties". Therein the chapters sort of touched some aspects of heterogeneities and has some results related to slip migration along the fault to long distances.

We did a detailed review about tremor and SSE models in the last two paragraphs of section 1 in the main text, we have now improved that review session and included Ray (2019) and Ray and Viesca (2017, 2019) as part of the reference. (Line 368-369 of tracked change version and Line 352-353 of clean version).

5. What parameters are considered to mean VW/VS ratio (line-379)?

Please see our response to comment #1 of reviewer #1 about VW/VS ratio.

6. The method section has lots of scope for improvements in terms of tight presentation. Many papers, mainly standard theoretical ones, are not cited. For example, consider citing Rice (1993) where radiation damping is introduced.

Thank you very much for your comment, we have re-worked on the method session to make the presentation more concise and added more references. (Line 326-370 of tracked change version and Line 315-354 of clean version).

7. Consider following corrections:

Correct for symbol inconsistency: D_{c} or D_c

D_c and other symbols have been checked throughout the text for consistency.

Line 51-52: "CSZ displays clear along-strike variation.....". Please remind along-strike variation of what?

Changed to "ETS in CSZ displays clear along-strike variation...". (Line 51 of tracked change version and Line 51 of clean version).

Line 137: typo corrected: heterogeneous.

Typos in L137, L342 and Figure 9 caption have been fixed.

Line: 273: please write all that is meant by 'etc.'

Removed ambiguous "etc."

Line 275-278: Please remind the readers about the scaling relations.

We have updated our manuscript to include a more detailed discussion about the scaling relation. (Line 289-290 of tracked change version and Line 421-434 of clean version).

Line 403-406: Please remind the readers the conventional seismological definition of tremor briefly. To what quantity the phrase 'a good first order approximation' referring to (also Line 72)? It may be suitable just to write slip rate of 1000Vpl is an indicator of tremor type situations.

Thank you for the suggestion. We have updated the manuscript to elaborate more about the conventional definition of tremor and our approach. (Line 464-467 of tracked change version and Line 283-284 of clean version).

For the detection threshold of tremor, please refer to our reply to question #2 of the reviewer #1.

References

- Audet, P., Bostock, M. G., Christensen, N. I., & Peacock, S. M. (2009). Seismic evidence for overpressured subducted oceanic crust and megathrust fault sealing. *Nature*, 457(7225), 76-78.
- Fagereng, Å. (2011). Fractal vein distributions within a fault-fracture mesh in an exhumed accretionary mélangé, Chrystalls Beach Complex, New Zealand. *Journal of Structural Geology*, 33(5), 918-927.
- Fagereng, Å., & Cooper, A. F. (2010). The metamorphic history of rocks buried, accreted and exhumed in an accretionary prism: an example from the Otago Schist, New Zealand. *Journal of Metamorphic Geology*, 28(9), 935-954.
- Fagereng, Å., & Sibson, R. H. (2010). Melange rheology and seismic style. *Geology*, 38(8), 751-754.
- Kao, H., Wang, K., Dragert, H., Kao, J. Y., & Rogers, G. (2010). Estimating seismic moment magnitude (M_w) of tremor bursts in northern Cascadia: Implications for the "seismic efficiency" of episodic tremor and slip. *Geophysical Research Letters*, 37(19).
- Wech, A. G., & Creager, K. C. (2011). A continuum of stress, strength and slip in the Cascadia subduction zone. *Nature Geoscience*, 4(9), 624-628.
- Yabe, S., & Ide, S. (2017). Slip behavior transitions of a heterogeneous linear fault. *Journal of Geophysical Research: Solid Earth*, 122(1), 387-410.
- Yabe, S., & Ide, S. (2018). Variations in precursory slip behavior resulting from frictional heterogeneity. *Progress in Earth and Planetary Science*, 5(1), 43.

REVIEWERS' COMMENTS

Reviewer #1 (Remarks to the Author):

The revised version of the manuscript entitled "Fault Zone Heterogeneities Explain Depth-dependent Pattern and Evolution of Slow Earthquakes in Cascadia" has adequately addressed all of my comments. Clarifications made in the main text and the additional supplementary figure have improve the discussions. I enjoyed reading the revised paper, and would recommend the paper for publication as is.

Reviewer #2 (Remarks to the Author):

I have gone through the revised version and the response to comments and I am satisfied with the changes and response. I have no further comment.

Pascal Audet

We sincerely thank all the reviewers and editor for their favorable consideration and valuable comments for our manuscript.

REVIEWER COMMENTS

Reviewer #1 (Remarks to the Author):

The revised version of the manuscript entitled "Fault Zone Heterogeneities Explain Depth-dependent Pattern and Evolution of Slow Earthquakes in Cascadia" has adequately addressed all of my comments. Clarifications made in the main text and the additional supplementary figure have improve the discussions. I enjoyed reading the revised paper, and would recommend the paper for publication as is.

We are thankful to Reviewer #1 for the second round of review and positive evaluation of our manuscript.

Reviewer #2 (Remarks to the Author):

I have gone through the revised version and the response to comments and I am satisfied with the changes and response. I have no further comment.

Pascal Audet

We are grateful to Dr. Pascal Audet for the second round of review and positive feedbacks.